# Coupling the regional climate MAR model with the ice sheet model PISM mitigates the melt-elevation positive feedback

Alison Delhasse [1], Johanna Beckmann [2,3], Christoph Kittel [4,1], and Xavier Fettweis [1]

[1]Laboratory of Climatology, Department of Geography, SPHERES, University of Liège, Liège, Belgium
[2]Climate Impact Research (PIK), Member of the Leibniz Association, Potsdam, Germany
[3]Securing Antarctica's Environmental Future, Monash University, School of Earth, Atmosphere and Environment, Clayton, Australia
[4]Institut des Géosciences de l'Environnement (IGE), Université Grenoble Alpes/CNRS/IRD/G-INP, Grenoble, France

**Abstract.** The Greenland Ice Sheet is a key contributor to sea level rise. By melting, the ice sheet thins, inducing higher surface melt due to lower surface elevations, accelerating the melt coming from global warming. This process is called the melt-elevation feedback and can be considered by using two types of models: atmospheric models, which can represent the surface mass balance, or SMB estimates resulting from simpler models such as positive degree day models. But these models generally use a fixed surface elevation. And on the other side, the ice sheet models represent the surface elevation evolution. The last ones do not represent the surface mass balance explicitly as well as polar-oriented climate models. A new coupling between the regional climate model MAR (Modèle Atmosphérique Régional) and the ice sheet model PISM (Parallel Ice Sheet Model) is presented here following the CESM2 (SSP5-8.5) scenario until 2100 at the MAR lateral boundaries. The coupling is extended to 2200 with a stabilised climate (+ 7 °C compared to 1961 – 1990) by randomly sampling the last 10 years of CESM2 to force MAR and reaches a sea level rise contribution of 64 cm. The fully coupled simulation is compared to a 1-way experiment where surface topography remains fixed in MAR. However, the surface mass balance is corrected for the melt-elevation feedback when interpolated on the PISM grid by using surface mass balance vertical gradients as a function of local elevation variations (offline correction). This method is often used to represent the melt-elevation feedback and prevents from a coupling too expensive in computation time. In the fully-coupled MAR simulation, the ice sheet morphology evolution (changing slope and reducing the orographic barrier) induces changes in local atmospheric patterns. More specifically, wind regimes are modified, as well as temperature lapse rates, which influence the melt rate through modification of sensible heat fluxes at the ice sheet margins. We highlight mitigation of the melt lapse rate on the margins by modifying the surface morphology. The lapse rates considered by the offline correction are no longer valid at the ice sheet margins. If used (1-way simulation), this correction implies an overestimation of the sea level rise contribution of 2.5 %. The mitigation of the melt lapse rate on the margins can only be corrected by using a full coupling between an ice-sheet model and an atmospheric model.

## 1 Introduction

The mass balance (MB) of the Greenland Ice sheet (GrIS) is a key factor of the future estimation of sea level rise (SLR) (Oppenheimer et al., 2019). Within the components of the GrIS MB, changes in Surface Mass Balance (SMB) and iceberg

discharge, surface meltwater runoff appears to be the main contributor to future SLR (Goelzer et al., 2013, 2020; Enderlin et al., 2014; Choi et al., 2021).

As a long-term consequence, the ongoing ice melt will result in a reduction of the ice sheet topography. Such a thinning will influence the regional atmospheric circulation, particularly affecting the spatial distribution of precipitation (Vizcaíno et al., 2010). Moreover, it will amplify surface melting through a positive feedback, since the lower ice sheet elevation results in intensified warming and increased melting (hereafter, melt-elevation feedback). Changes in topography are generally not considered in climate models, but they are in Ice Sheet Models (ISMs). Conversely, atmospheric models, particularly Regional Climate Models (RCMs), can represent the SMB, and its components in a more realistic way than ISMs by explicitly resolving the physical polar processes involved in interactions between the atmosphere and ice or snow surfaces. Therefore, the most optimal method to represent the melt-elevation feedback would involve coupling a RCM with an ISM.

Yet, this kind of coupling present two main challenges. First, a RCM-ISM coupling is nontrivial due to disparities in spatial and temporal scales between the two model types. Simulations of ice flow processes require relatively large time steps (on the order of one month), in contrast to atmospheric dynamic resolution (on the order of one minute) which is significantly shorter. Conversely, ISMs are implemented on a finer grid (on the order of one kilometre) compared to RCMs (on the order of ten kilometres). These differences result in extended computational times for coupled simulations. Furthermore, depending on the specific ISM used in a RCM-ISM coupling, the response to climate change may significantly vary (Goelzer et al., 2013, 2020), while RCMs tend to simulate relatively similar SMB for the same large-scale forcing (Fettweis et al., 2020). This implies the necessity for multiple couplings with several ISMs and RCMs to accurately quantify uncertainties.

An often-used alternative to coupling is to impose atmospheric conditions from an RCM onto an ISM. In such cases, either the ISM calculates SMB based on the monthly mean temperature change (Robinson et al., 2011) or the SMB is directly imposed on the ISM (Goelzer et al., 2013). However, as the topography and ice mask remain fixed in the RCM, this fails to account for the positive feedback between melt and elevation. In this context, it is possible to employ vertical gradients as a function of local elevation variations (implicitly considering the melt-elevation feedback) to correct the elevation-dependent outputs generated by the RCM for topographic variations simulated by the ISM (Franco et al., 2012; Helsen et al., 2012). Subsequently, ISMs can employ the corrected outputs from the RCM as direct inputs. This offline method has notably been employed in the Ice Sheet Model Intercomparison Project for CMIP6 (ISMIP6, Goelzer et al., 2020) exercise. The use of an offline correction is effective as long as SMB (especially melt) is predominantly influenced by surface elevation changes through the temperature lapse rate. However, its effectiveness may diminish when surface elevation changes start to impact synoptic circulation patterns, leading to alterations in precipitation patterns, for example. Both approaches were compared in Le clec'h et al. (2019) using the GRISLI (Grenoble Ice Sheet and Land Ice, an ISM) and MAR (Modèle Atmosphérique Régional, a RCM) models, driven by MIROC5 (from CMIP5). The study underscored the need for coupling beyond 2100 due to the melt-elevation feedback and changes in precipitation-circulation changes that cannot be accurately represented when using a simple offline correction, especially as topographic changes become increasingly substantial.

Given that the coupling depends on the chosen ISM, we introduce a new coupling between the climate model MAR and the Parallel Ice Sheet Model (PISM), following an extremely warm scenario (CESM2 SSP5-8.5) until 2200. This coupling

explicitly accounts for the melt-elevation feedback and is compared against an alternative offline correction method for SMB. We have three main objectives: (1) to analyse the evolution of the GrIS until 2200 under an extreme scenario using this new coupling, (2) to evaluate the ability of the offline method to represent the melt-elevation feedback by comparing our coupled simulation with the one-way experiment, and (3) to elucidate which atmospheric feedback mechanisms and physical processes are influenced by changes in surface topography.

Section 2 describes the methodology and different experiments used in the study. The first part of the result in Section 3.1 presents the GrIS evolution as simulated with our coupling until 2200. Section 3.2 compares 1-way and 2-way experiments by evidencing new negative feedback triggered by the evolving surface topography of the GrIS in the 2-way-coupling method. Results are discussed in Section 4 and we end by the conclusion in Section 5.

## 2 Data and methodology

### 2.1 Models

#### 2.1.1 Regional climate model MAR

The MAR model is a hydrostatic regional climate model specially designed to represent polar climates. It is widely utilised over Greenland (Delhasse et al., 2020; Fettweis et al., 2020; Hofer et al., 2020; Fettweis et al., 2021) and extends its application to Antarctica as well (Agosta et al., 2019; Kittel et al., 2020; Amory et al., 2021). The version 3.11.5 of MAR (MAR hereafter Fettweis et al., 2021) is used here at a spatial resolution of 25 km. For the coupling process, the important variables are SMB and surface temperature (ST), which are required as inputs to the ISM. These surface variables originate from the interactions between the atmosphere and the first snow/ice layers of the ice sheet, which are represented by the Surface-Ice-Snow-Vegetation-Atmosphere-Transfer (SISVAT) module within MAR. The ability of MAR in simulating atmosphere/ice interactions, particularly near-surface temperature (Delhasse et al., 2020), plays a key role in the successful coupling, as it determines the two input variables required by the ISM (SMB and ST). Note that the MAR evaluation mentioned here is based on a version of MAR at a spatial resolution of 15 km (Delhasse et al., 2020).

As MAR is a regional climate model, it necessitates lateral forcing fields every 6 hours to accurately represent its own climate within the chosen domain. Therefore, we selected one of the most climate-sensitive models (Hofer et al., 2020) from the CMIP6 model ensemble (CESM2) using the SSP5-8.5 scenario as outlined by the IPCC (Eyring et al., 2016; O'Neill et al., 2016; Riahi et al., 2017), which was available at the offset of our study. The equilibrium climate sensitivity (a hypothetical value of global warming at equilibrium for a doubling of atmospheric $CO_2$ concentration relative to a 140-year period in the pre-industrial) of CESM2 is + 5.2 °C (mean of CMIP6: + 3.2 °C, Meehl et al., 2020). This choice was motivated by our intention to represent the most extreme (i.e. warmer) future scenario for the GrIS. SSP5-8.5 represents the most extreme scenario with an additional radiative forcing of 8.5 $\mathrm{Wm^{-2}}$ projected for 2100. The aim is to highlight the limitations of both methods in representing the melt-elevation feedback under conditions of extreme warming.

CESM2 (SSP5-8.5, 6-hourly outputs) is currently only available until 2100. To extend our simulations up to 2200, we force MAR with the last 10 years (2091–2100) of CESM2 data, randomly sampled across the period of 2101–2200 (Table S1 in the Supplement). This means we apply a stabilised climate (mean conditions and interannual variability) to Greenland over 100 years. This extension of the large-scale forcing enables us to distinguish the effect of the rapidly increased warming projected with this scenario until 2100 compared to the effect of continued stable warming of $+7\,^{\circ}\text{C}$ above Greenland until 2200.

The polar version of MAR requires a fairly long spinup period to reach an equilibrium state for both the snowpack and the atmosphere. Concerning the snowpack, the parameters that are important for achieving an equilibrium state and representing a coherent configuration (temperature, density and liquid water content, Lefebre et al., 2003) are pre-initialised based on former simulations. These simulations have undergone an extensive spinup process spanning over 50 years to establish a coherent representation of the snowpack (Fettweis et al., 2020).

## 2.1.2 Ice Sheet Model PISM

To represent the dynamics and surface elevation of the Greenland Ice Sheet (GrIS), we utilise the Parallel Ice Sheet Model (PISMv1.2.1, called PISM hereafter), a high-resolution numerical ice-sheet/ice-shelf model (Bueler and Brown, 2009; Winkelmann et al., 2011). In PISM, the geometry, temperature, and basal strength of the ice sheet are incorporated into stress balance equations at each time step to determine the ice velocity. In some models, the full stress field is calculated by using the full

Stokes equation. But this is computationally expensive. As an ice sheet can be treated as "shallow" (meaning the area of the ice sheet is far greater than its thickness), PISM employs two approximations for shallow ice sheets: the Shallow Ice Approximation (SIA) and the Shallow Shelf Approximation (SSA).

The SIA simplifies by neglecting membrane stresses, which involve along-flow stretching and compression, as well as transverse stresses, which result from lateral drag against slower ice or valley walls. The viscosity of the ice, affecting its

flow velocity, is modulated by an enhancement factor E. We set $E = 3$ in our experiments, a value typically used for the GrIS (Aschwanden et al., 2013; Beckmann and Winkelmann, 2023). A typical SIA velocity profile in a cross-section shows zero velocity at the bed (frozen to the bed) and increasing velocities at the surface.

Faster flowing ice, such as ice streams, glaciers, and shelves, is typically approximated using the SSA. In this case, longitudinal stretching dominates, and membrane and transverse stresses must be taken into account. The ice base is assumed to be

slippery, and the full ice column moved at the same horizontal speed. This plug flow allows for depth averaging in the SSA equation. While SIA can be numerically solved individually in each ice column, the SSA is nonlocal, meaning the velocity of a certain grid point depends on the whole spatially distributed stress field.

PISM combines both approximations into a hybrid stress balance mode (Bueler and Brown, 2009; Aschwanden et al., 2012; Winkelmann et al., 2011). Throughout the entire domain, PISM calculates velocities for both SIA and SSA. SSA velocities

result in very small velocities in the ice interior, where membrane stresses are low and basal resistance is high. They increase in regions with basal slip. The overall velocities in PISM for grounded ice are the sum of SSA velocities and SIA velocities,

expressed as Eq. 1 (Winkelmann et al., 2011). This superposition method helps avoid discontinuities in the model.

$$v = v\_SIA + v\_SSA \tag{1}$$

Basal sliding of the ice over the bedrock introduces basal resistance. The speed of basal sliding is determined by the sliding law, typically a power law relating to the basal shear stress ($\tau_b$) and yield stress ($\tau_c$). We use an exponent of $q = 0.6$ in our "pseudo plastic" sliding law (Eq. 2).

$$\tau_b = -\tau_c \frac{u}{u_{threshold}^q |u|^{1-q}} \tag{2}$$

To determine the yield stress $\tau_c$ we use the mohr-coulomb criterion in PISM. The model considers basal resistance based on the hypothesis that the ice sheet rests on a till layer. The yield stress represents the strength of this aggregate material at the base of an ice sheet. When yield stress is lower than the driving stress ($\tau_c < \tau_d$) there is likely to be sliding, and thus faster velocities can be observed. The driving stress in turn is dependent on the ice thickness ($H$) and surface gradients ($h_s$) of the ice: $\tau_d \propto H h_s$. The thicker and steeper the ice, the higher the driving stress and most probably the ice velocity.

The properties of the till are further approximated by using material properties such as the friction angle. We vary the till friction angle linearly between $5°$ and $40°$ with respect to bedrock elevation (between -700 m and 700 m), following Aschwanden et al. (2016). This variation in friction angle leads to lower friction at lower altitudes and below sea level, resulting in increased surface velocities at the margins of the ice sheet, thus improving the match of flow structure for the glaciers.

To match the present-day extent of the ice sheet, we impose a strong negative surface mass balance at the margins of the Greenland present-day ice mask. This setup allows only for ice retreat in our experiments.

We also enforce a minimum thickness of 50 m for floating ice at the calving front and utilise the von Mises calving law, which is suitable for glaciers in Greenland (Morlighem et al., 2016).The von Mises yield criterion is a widely adopted yield criterion in the fields of solid mechanics and structural analysis. Calving is predominantly influenced by stretching, and the von Mises stress is a fundamental measure for quantifying deformation and fracture. Therefore, it directly impacts the calving speed and is incorporated in PISM following Eq. 3.

$$c = ||v|| \frac{\sigma}{\sigma_{max}} \tag{3}$$

where $||v||$ is the velocity perpendicular to the ice front, $\sigma$ is the von Mises stress for ice (Morlighem et al., 2016), and $\sigma_{max}$ is a threshold. If the von Mises stress is greater than the threshold the ice front retreats ($c > ||v||$) if it is smaller the ice front advances. PISM uses a threshold value of $1x10^6$ Pa.

All other parameters are set to default values (University of Alaska Fairbanks, 2019). Our simulations do not consider bedrock deformation or changes in ice-ocean interaction, as we maintain constant submarine melt rates.

### 2.1.3 Inisialisation of PISM

PISM is forced by yearly ST and SMB from MAR forced by CESM2. To achieve a stable spinup state, we forced PISM with the MAR mean fields (ST and SMB) over $1961-1990$, when the GrIS was close to balance (Fettweis et al., 2017). However,

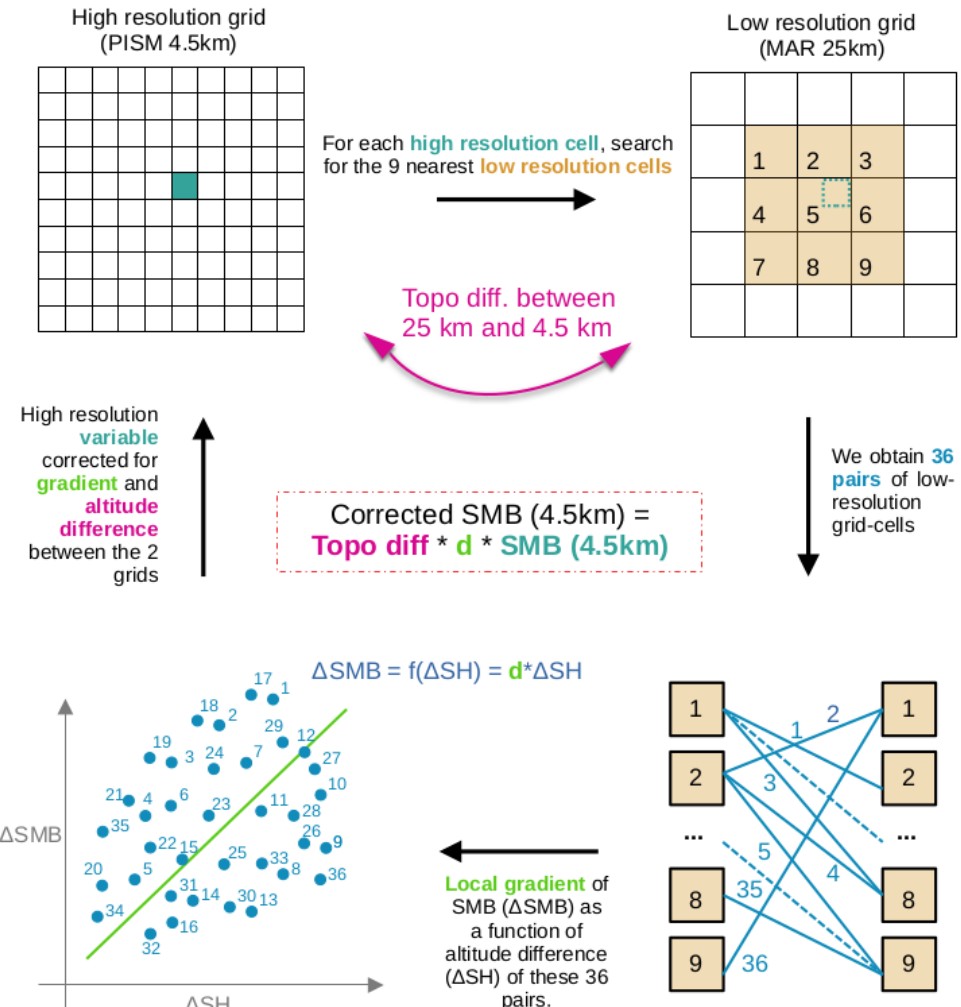

**Figure 1.** Steps of the offline correction as described in (Franco et al., 2012). After interpolation of a variable (SMB, surface mass balance, in this figure) from a low to higher resolution grid, this variable is corrected to consider the influence of the temperature lapse rate with altitude. The correction is based on a local gradient (d) calculated by considering SMB differences ($\Delta$SMB) between 9 nearest low-resolution grid cells in the neighbourhood of the high-resolution grid cell position in function of the surface elevation difference ($\Delta$SH). Modified from Wyard (2015).

for a realistic thermodynamics representation of the ice sheet, the temperature evolution of the last glacial cycle has to be considered, because the surface temperature slowly propagates down the ice column and determines the vertical ice profile of

the ice sheet. The ice profile determines the ice softness and deformability, thus affecting the flow velocity of the ice.

     For a glacial spinup, we assume that the initial state of the ice sheet prior to a glacial cycle is identical to the present-day state, including ice topography and surface temperatures. Therefore, we start with a contemporary ice sheet and force it with surface

temperatures corresponding to the last glacial cycle. To maintain model continuity, historic surface temperatures spanning the last 125 000 years were incorporated as climate anomalies into the present-day climatological mean (ST for 1961 – 1990). This
approach means that temperature anomalies were zero at both 125 000 years ago and at the present day, but they varied during the glacial period. As our coupled spinup progresses, we obtain different surface topographies that result in varying surface temperatures and, consequently, distinct climatological mean values (Section 2.3.1). By using these anomalies, we ensure that the assumption of equivalent glacial states before and after the glacial cycle remains valid, as the anomalies are consistently zero at those two time points.

The first model initialisation (Fig. 2) spanned 125 000 years,incorporating a scalar temperature anomaly derived from the 2D-temperature mean field of 1961 – 1990, a period when Greenland was near a state of balance. This 2D temperature and SMB mean field were calculated by MAR using the present-day PISM topography. The historical time series (Johnson et al., 2019) includes the temperature derived from Oxygen Isotope Records from the Greenland Ice Core Project (GRIP, Johnson et al., 2019). To optimise computational efficiency, we followed the grid refinement defined by Aschwanden et al. (2016). Starting
in SIA-only mode, and an 18 km grid at -125 000 years, we refined our grid to 9 km at -25 000 years, and to 4.5 km at -5 000 years. For the last -1 000 years, we maintained a fixed resolution but introduced SSA to the SIA stress regime to represent the behavior of fast-flowing outlet glaciers. Note that the initialisation of PISM ends after the reference period 1961-1990 when the ice sheet is assumed to have been in a quasi-equilibrium.

## 2.2 Coupling method

The coupling between both models has been performed by exchanging yearly outputs (specifically, SMB and ST from MAR, and ice thickness from PISM) on the 1st January of each year from 1991 to 2200 as described in Le clec'h et al. (2019). For MAR, this induces updating the surface elevation and ice extent of the ice sheet at the beginning of each year with PISM results from the previous year, whereas SMB and ST are used as forcing fields for PISM.

Before any data exchange between the models, data has to be interpolated onto the destination grid because the two models
were run at two different spatial resolutions (25 vs 4.5 km). The surface elevation simulated by PISM is then aggregated on the MAR grid at 25 km using a four-nearest-neighbour distance-weighted method. Conversely, MAR variables undergo interpolation onto the PISM grid at 4.5 km using the same method. However, a further correction is applied to consider the difference in altitude between the two grids at the time of interpolation thanks to local vertical SMB/ST gradients. This method is described in Franco et al. (2012) and is called offline correction hereafter.

Firstly, a linear and elevation-dependent gradient (Fig. 1) is calculated over the MAR grid, considering the values of the considered variable (for instance, SMB at 4.5 km, as illustrated in Fig. 1) of the nine surrounding grid cells of the corresponding-position in the high-resolution grid. This gradient is specific to each PISM grid cell and is determined locally. An example of such a gradient is available in Fig. S1 in the Supplement. Subsequently, these gradients are utilised to correct the variable during its interpolation onto the PISM grid. The correction is performed by multiplying the interpolated variable by the gradient and
the difference in surface elevation between the grid cells in MAR and in PISM. This offline correction is specifically employed to correct variables that are influenced by temperature lapse rate with altitude, namely temperature and derived variables.

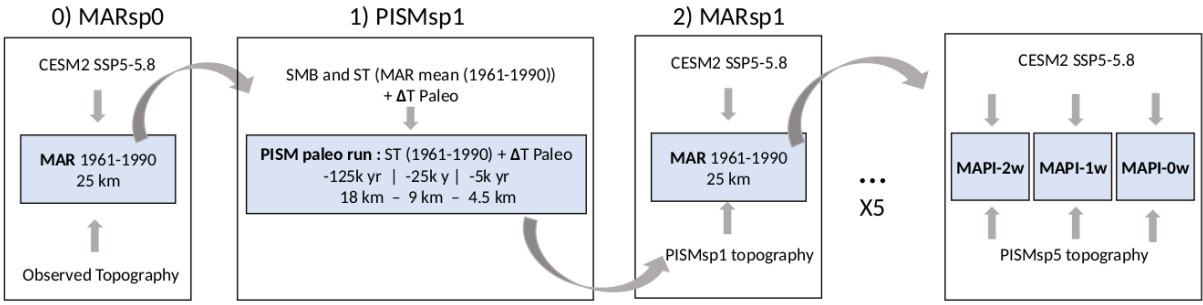

**Figure 2.** Steps of the coupling initialisation. Each MAR step corresponds to a 30-year long run over the reference period (1961-1990). And each PISM step consists of a new initialisation cycle of PISM as described in Section 2.3.1.

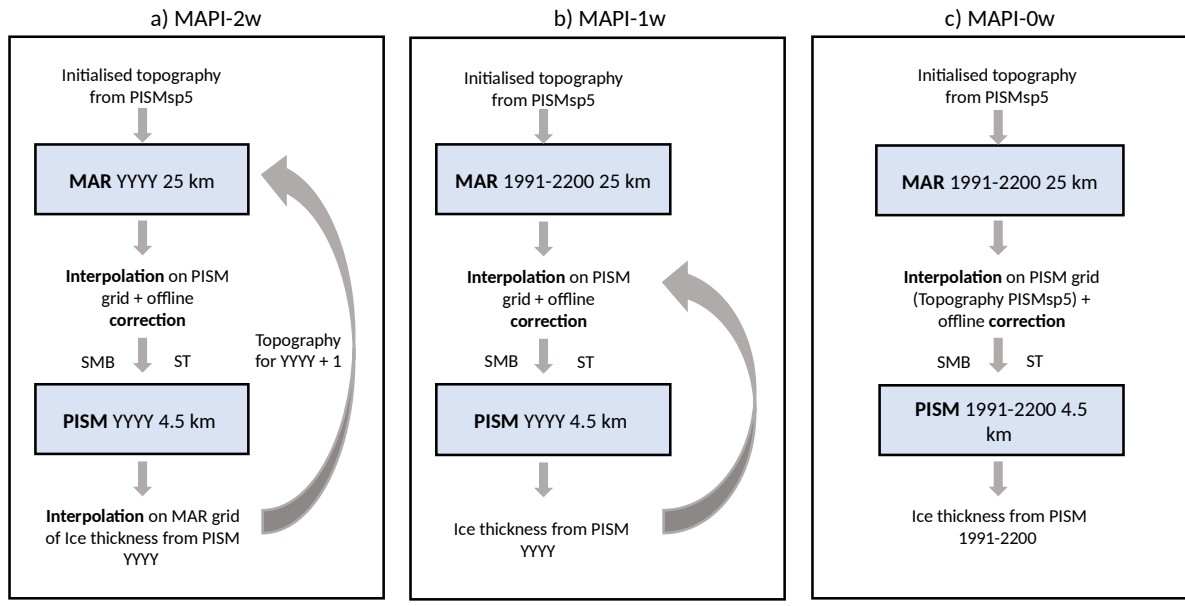

**Figure 3.** Coupling details for a) 2-way, b) 1-way and c) 0-way experiments.

## 2.3 Simulations

### 2.3.1 Initialisation of the coupling

The coupling requires initialisation to achieve an equilibrium state between the two models over a reference period (1961 – 1990). Successive forcings of each model by the other one should produce similar results to the previous iteration over the

same reference period (Fig. 2). These simulations are called spinup hereafter. In practical terms, we firstly forced PISM using the SMB and ST climatology of MAR 1961 – 1990 (MARsp0, based on the observed topography and ice mask, and CESM2 as a large-scale forcing field, Howat et al., 2014, 2017) and temperature anomalies of the last glacial cycle (see Section 2.1.3) resulting in a first equilibrium state (PISMsp1). This method assumes that the ice sheet topography before the last glacial cycle was similar to the preindustrial one and reiterates this process should correct for errors in ice thickness. The next step (MARsp1) consists in running MAR using the new ice extent and topography from PISMsp1 over the same period (still 1961 – 1990). The corrected surface topography (PISMsp1) together with the corrected SMB and ST climatology (MARsp1) are the new base to re-start our initialisation over a whole glacial cycle, as described in Section 2.1.3. The new surface topography, PISMsp2, is then used to derive the new climatological mean field of (1961 – 1990) with MAR (MARsp2). We repeated these successive forcings (5 iterations are needed here) until both models reached an equilibrium state regardless of the new forcing. This means that differences between the two spinup stages no longer influence the other model (Fig. S2 in the Supplement).

The PISMsp5 topography, the last step of the initialisation process, will be the initial state of the different simulations compared here and is used to run the MAR reference simulation over the reference period (MARref). As our projections could not be evaluated, we evaluated the performances of MARref over the present. To do so, we compared MAR results over the current period (1961 – 1990), with the initialised topography (PISMsp5) forced on one hand, by the Earth System Model used for projections (CESM2) and on the other hand ERA5 reanalysis (Hersbach et al., 2020), considered as observations and well representing the current climate. The main point of this evaluation is that MARref is significantly colder than MAR forced by ERA5 in the south of Greenland, but this bias does not significantly influence SMB results (See Fig. S3 in the Supplement). It is due to the cold bias of CESM2 compared to reanalyses (Hofer et al., 2020).

The ice sheet topography and velocity field of the PISMsp5 final run and their difference to observational data sets are depicted in the Supplement (Fig. S4). The ice sheet thickness of the final spinup is overestimated up to 150 m in the northeast and southwest of the GrIS. In comparison, the northwest and central west are underestimated up to 200 m compared to the observational data set (Fig. S4a). As there is no complete observational velocity data set from 1961 to 1990, we, therefore, compare it with the complete velocity data set by Joughin et al. (2018), which gives the average velocities from 1995 to 2015. Our comparison shows a general agreement of the velocity pattern with an average difference between modelled and observed ice speeds of $\pm\, 80\,\mathrm{m\,yr^{-1}}$ on the margins (Fig. S4b). In some fast-flowing glacier regions, differences are well larger. However, the coarse resolution (4.5 km) compared to the proximity of smaller glaciers (500 m), which are solved by the observations, lead to strong deviation in their comparison. Furthermore, from 1995 to 2015, Greenland was not in balance, and glaciers were already experiencing speed up and retreat (King et al., 2020).

### 2.3.2 Coupled simulation

The first simulation is the 2-way experiment which consists of a coupled simulation of MAR and PISM called MAPI-2w hereafter. We started to run MAR in 1991 with the PISMsp5 topography forced with CESM2 (Fig. 3a). At the end of this first year, we interpolated SMB and ST on the PISM grid with the offline correction. Then, PISM is running for the same year and produces a new ice thickness that will further be aggregated onto the MAR grid to start the following year (i.e. 1992) with an

updated topography. When the MAR topography is updated, we also update the ice mask in function of PISM ice extent. The melt-elevation feedback is, therefore, explicitly taken into account by the MAPI-2w simulation through an evolving topography in MAR.

### 2.3.3 Uncoupled simulations

The 1-way experiment (called MAPI-1w hereafter) consists in a simulation where MAR is running with PISMsp5 topography (built over the reference period) for $1991 - 2200$ without any more interaction from PISM to MAR (Fig. 3b). Then we interpolated the yearly results of SMB and ST from MAR to the PISM grid with the offline correction. Thus, the new PISM input variables were corrected for changes in the surface height of the evolving ice sheet topography of PISM compared to the fixed MAR surface elevation. The MAPI-1w experiment considers then the melt-elevation feedback a posteriori through the offline correction, meaning that it is not explicitly solved by any of the two models (MAR or PISM), nor are the implied physical processes. As MAR is evolving alone, no update on the ice mask has been done. To be consistent, we consider the smallest ice mask all along the analyses, meaning the one in 2200 of the MAPI-2w simulation, to compare both simulations.

We also consider a PISM simulation forced with MAR-fixed-topography corrected over the initial PISM topography (PISMsp5). This experiment is called the MAPI-0w experiment (Fig. 3c) hereafter due to its non-consideration of the melt-elevation feedback.

## 2.4 Representation of the results

The coupling aims to estimate the total MB of the GrIS by directly simulating the dynamical components with PISM and employing the SMB components, as simulated by MAR, as forcing for PISM.

For the sake of consistency of the results, we decided to present all results on the PISM grid, whether they are PISM or MAR outputs. The MAR variables used in the analyses below are therefore interpolated on the PISM grid. While those from the coupled simulation explicitly include the influence of the melt-elevation feedback, the variables from the uncoupled simulation (MAPI-1w) are corrected offline during the interpolation. This correction is applied to the variables dependent on the surface elevation influence, i.e., temperature, SMB, meltwater production and runoff. On the other hand, the following variables will not be corrected during their interpolation since they do not depend on the evolution of the surface elevation: total precipitation (snowfall and rainfall) and wind. However, some comparisons have been carried out on the MAR grid, but this is well specified each time.

The two main PISM simulations, which are compared hereafter (MAPI-2w and -1w), evolve independently and consequently have differences in surface topography. These differences are only responsible for 10 % of the MB differences between two experiments in 2200 (Fig. S6 in the Supplement) and will be further neglected. Throughout the analysis, we will consider the MAR results interpolated on the MAPI-2w coupled PISM grid (4.5 km) regardless if they are from MAR coupled or uncoupled simulation.

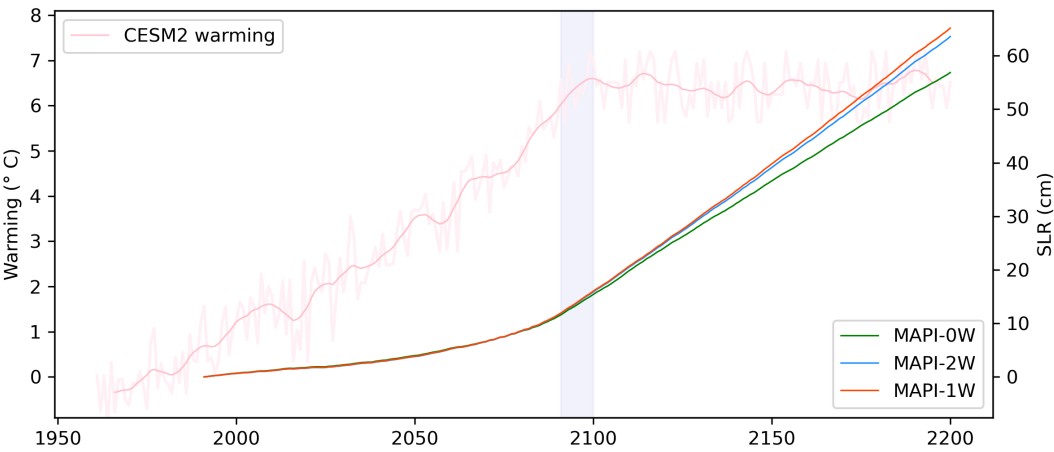

**Figure 4.** Contribution to sea level rise (SLR, cm) of the Greenland Ice Sheet according to MAR-PISM 2-way (in blue),) 1-way (in red) and 0-way (in green) experiments. In pink is the corresponding warming (°C) applied in the MAR lateral boundaries following CESM2 SSP5-8.5 (mean temperature at 600 hPa over Greenland). The last 10 years of CESM2 randomly sampled until 2200 to extend the CESM2-forcing of MAR are in grey.

## 3 Results

### 3.1 Coupled MB and SMB

This section is dedicated to describing future changes in total mass balance of the GrIS, surface mass balance and its components compared to the reference period as simulated with the 2w-coupling experiment.

Our findings indicate a rapid increase in annual mass loss under an extreme warming scenario (Fig. 4). By 2100, this corresponds to a contribution of 16 cm to sea level rise (equivalent to a total ice mass loss of more than 50 x$10^3$ Gt) due to a global temperature increase of + 7 °C (+ 6.8 °C on average for 2091 – 2100) compared to our reference period (1961 – 1990). Beyond 2100, as the temperature stabilises at + 7 °C (on average for 2101-2200), the ice sheet continues to lose mass, resulting in a sea level rise contribution of 64 cm since 1991 (equivalent to a total ice mass loss of more than 200 x$10^3$ Gt). Despite

the stabilisation of warming after 2100, the mass loss continues to rise until 2200 due to earlier warming-induced acceleration before 2100.

    While the GrIS is experiencing a retreat of several kilometers along its periphery (see Fig. 5), the cumulated mass balance (MB) becomes markedly negative by 2200, resulting in a decrease in surface elevation of several hundred metres along the margins of the GrIS. Notably, the western margin is particularly impacted by this mass loss and the subsequent elevation

decrease. By 2200, many peripheral glaciers seem to disappear, especially in the east and north of the island.

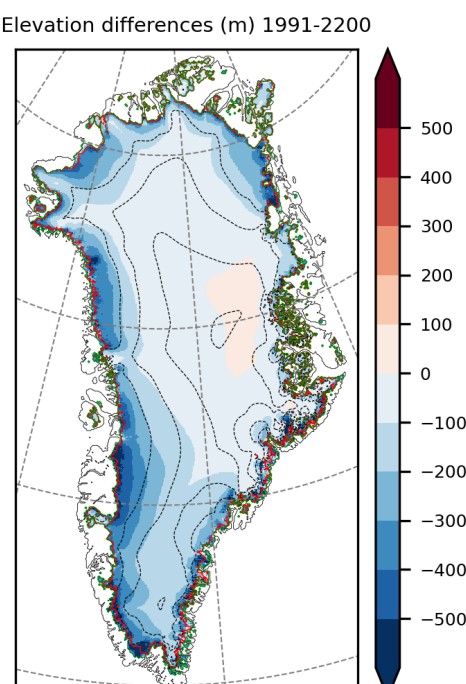

**Figure 5.** Surface elevation changes (m) as simulated by the 2-way coupling between MAR and PISM between 1991 and 2200. In green, the ice extent as in 1991 and in red as in 2200.

The SMB decrease largely drives the increase in mass loss. To attribute the specific contributors to SMB loss, we examine the native 25 km output MAR-2w (Fig. 6 solid lines). The SMB evolution is characterised by a sharp decrease from 2050 to 2100, followed by a slowdown from 2100 to 2200 as the climate stabilises again. These changes in SMB are predominantly attributed to runoff (RU hereafter) resulting from increased meltwater production (ME hereafter), which does not refreeze into
the snowpack.

Due to global warming, we expect higher precipitation rates, especially liquid precipitation due to the lower surface elevation. The snowfall (SF) evolution remains constant throughout the simulation period (Fig. 6). The slight increase in total precipitation is mainly due to increased rainfall (RF). Interestingly, it is worth noting that only a small part of the RF increase (approximately 1 % when comparing RF from MAR coupled, solid lines, and uncoupled, dashed lines) can be attributed to the reduction in
surface elevation, which leads to the conversion of more SF into RF.

The spatial distribution of the SMB component changes is mainly explained by the warming scenario, emphasised by the decreasing surface elevation until 2200 (Fig. 7a). Both ME and RU (Fig. 7e and f), which drive the spatial pattern of the SMB (Fig. 7d), are projected to occur further inland. For instance, by 2200, almost the entire southern half of Greenland is affected by RU in 2200. This leads to a decrease in SMB in these regions, subsequently triggering changes in surface topography,
evidencing the dependence of the MB on the SMB.

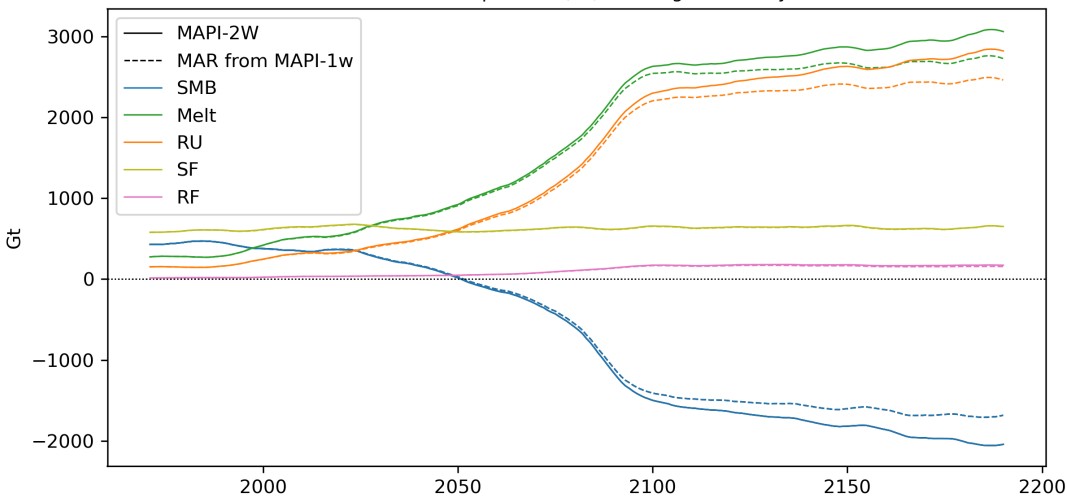

**Figure 6.** Surface mass balance (SMB, in blue), Meltwater production (ME, in green), meltwater Runoff (RU, in orange), Snowfall (SF in yellow) and Rainfall (RF, in pink) evolution (in Gt) as simulated by MAR 2-way coupled with PISM (MAPI-2w, solid lines) from 1991 to 2200. Dotted lines are corresponding evolution as simulated by MAR uncoupled (MAPI-1w).

While total precipitation over Greenland does not change significantly (Fig. 6), its spatial distribution does change with topography compared to the reference period in the coupled simulation. There is a significant decrease in total precipitation (SF + RF, Fig. 7c) over the southeast due to synoptic features of the large-scale forcing (CESM2, not shown). Conversely, our simulation projects a significant increase over the west and north of Greenland. The westward increase can be attributed to ice sheet thinning, which allows clouds to penetrate further inland due to reduced topographic barrier effect and a delayed condensation due to further lift-up of air masses. Additionally, a synoptic pattern originating from the CESM2 forcing contributes to this precipitation increase (not shown). The changes in snowfall over the north of Greenland are attributed to higher humidity content associated with atmospheric warming, as this region is typically dry and cold under present-day conditions. This results in a slight overall increase in precipitation over the entire ice sheet, primarily due to an increas in RF resulting from global warming and surface elevation changes, as explained previously. However, these significant alterations in precipitation spatial distribution do not substantially influence the overall SMB pattern, as runoff changes outweigh them.

The reduction in mass loss from the ice sheet is accompanied by an overall speedup of the ice dynamics further inland and a slow down at the margins (Fig. 8a). Surface velocity is directly linked to the driving stress. Subsequently the spatial pattern of changes in driving stress (Fig. 8b) mainly explains the spatial pattern of changes in surface velocities. The driving stress depends on the product of ice thickness and surface slope. Notably, the pronounced thinning occurring at the ice sheet margins explains the reduction in driving stress in these regions. While there is also an increase in surface slope at the margins,

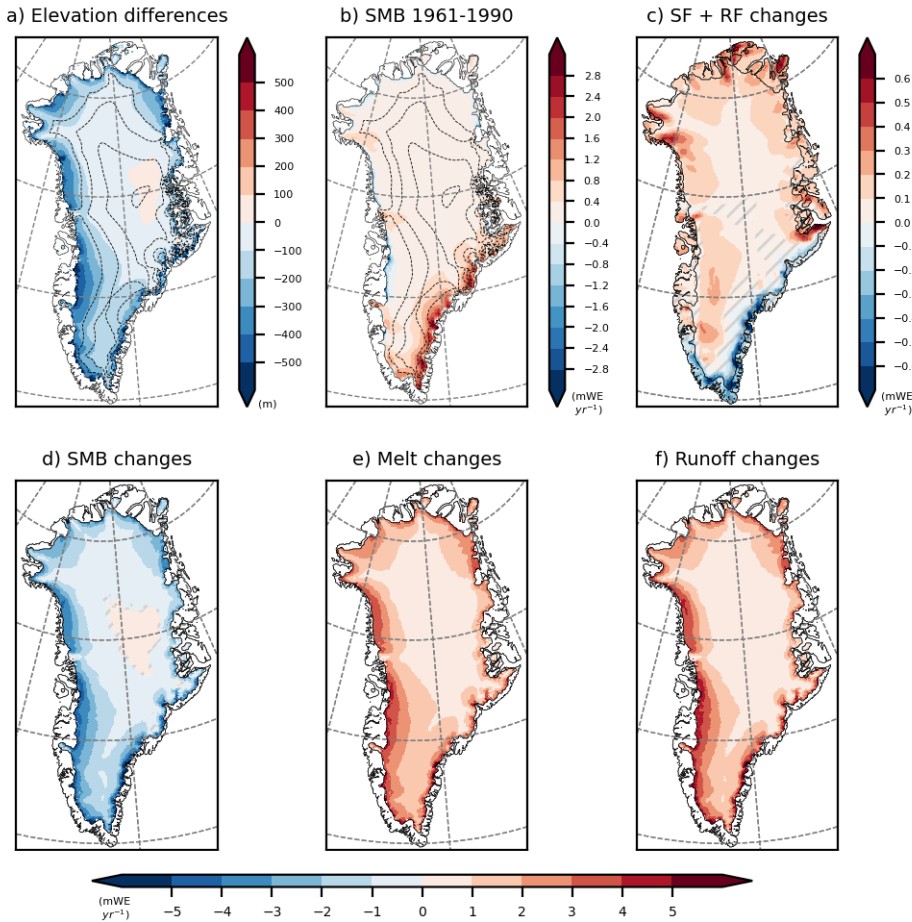

**Figure 7.** a) Surface elevation changes (m) between 1991 and 2200 as simulated by the MAR-PISM 2-way coupling (MAPI-2w). b) Surface mass balance (SMB, mWE yr$^{-1}$) for the reference period (1961 – 1990) as simulated by MAPI-2w. c) Precipitation (snowfall, SF and rainfall, RF, mWE yr$^{-1}$), d) SMB (mWE yr$^{-1}$), e) Meltwater production (mWE yr$^{-1}$) and f) meltwater Runoff (mWE yr$^{-1}$) changes in 2171 – 2200 compared to the reference period. Non-significant changes are hatched (smaller than the interannual variability of the reference period).

which would typically increase the driving stress, the thinning effect holds greater significance and determines the reduction in driving stress. In contrast, although less pronounced than at the margins, the ice interior still experiences an increase in surface slope. Further inland, the amplified surface slope emphasises the driving stress, especially as the thinning is smaller than at the margins. Consequently, the increased driving stress leads to higher surface velocities.

The overall pattern of speedup in the ice interior and slow down at the margins is observed in both the 2-way and 1-way experiments. However, in MAPI-2w, ice thickness is slightly larger at the margins and thinner in the ice interior than in MAPI-1w. At the margins, this result in a reduced surface slope and, when compared to the 1-way experiment, leads to slower velocities (Fig. S5b in the Supplement).

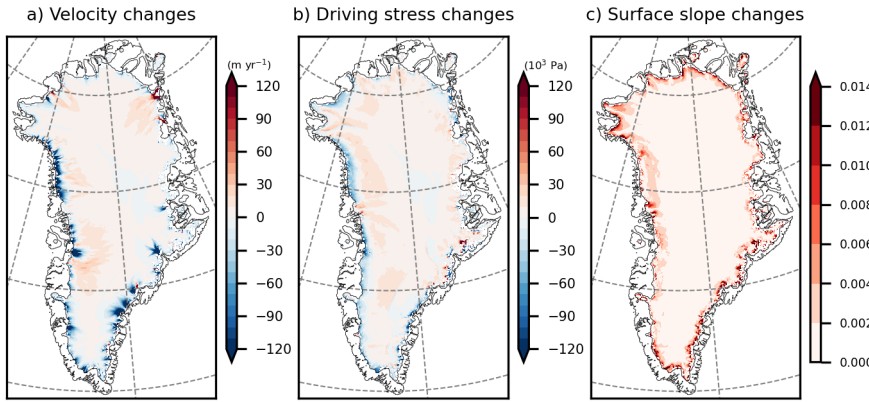

**Figure 8.** Changes (2200 – 1990) in a) velocity (m yr$^{-1}$), b) driving stress ($10^3$ Pa) and c) surface slope of the coupled MAR-PISM simulation (MAPI-2w).

## 3.2 Comparison of coupled and uncoupled experiments

This section focuses on the differences in the results obtained from the two approaches of considering the melt-elevation feedback (MAPI-1w vs MAPI-2w).

When comparing the total mass loss in 2200 (Fig. 4), the two strategies for representing the melt-elevation feedback (MAPI-1w vs MAPI-2w) do not lead to significantly different total mass losses in 2200. Specifically MAPI-2w result in a total ice loss of 229 x10$^3$ Gt, while MAPI-1w result in 234 x10$^3$ Gt. This means that MAPI-1w overestimates the SLR contribution by 2.5 % (equivalent to 1.6 cm) compared to MAPI-2w. In contrast, MAPI-0w largely underestimates ice loss by 10.5 % (equivalent to 6.7 cm less of SLR contribution than MAPI-2w) due to its non-representation of the melt-elevation feedback. These discrepancies become more pronounced as the climate stabilises.

The overestimation of MB by MAPI-1w could be contrary to the intermediate results from MAR before interpolation and correction onto PISM-grid of both MAPI-1w and -2w simulations. If we look at these results (raw MAR outputs) before interpolation and PISM forcing, the melt rate outputs in the fully-coupled mode (Fig. 6 solid lines) are higher than in the one-way coupled simulation (Fig. 6 dashed lines). However, after interpolation, when the MAR results are onto the PISM grid and subsequently used to force PISM, MAPI-1w gave higher melt rates than MAPI-2w (Fig. 9a-c). This discrepancy highlights that when we apply the offline correction to correct the SMB from the melt-elevation feedback (MAPI-1w on PISM grid), the SMB becomes excessively negative compared to the 2-way coupling, which explicitly accounts for this feedback. This finding is contrary to (Le clec'h et al., 2019), as discussed later in this study.

The corrected SMB provided to PISM in both MAPI-1w and MAPI-2w differ significantly at the ice sheet margin (Fig. 9a), indicating a greater mass loss for MAPI-1w. The different SMB components are analysed here (Fig. 9) to identify what is the cause of this underestimation in the corrected SMB for MAPI-1w. Firstly, whether on the east or west coast, there is a different distribution of total precipitation when simulated by MAPI-2w compared to MAPI-1w (Fig. 9f). Precipitation falls

further inland (positive differences) in the coupled mode due to the flattened slope, although this difference is not significant compared to annual variability (standard deviation for 2171 – 2200). Therefore, this does not explain the SMB differences in the margins between the two simulations. The main driver is the meltwater runoff (Fig. 9b), resulting from the excess of ME in MAPI-1w not refreeze in the snowpack (Fig. 9e). ME depends on sensible heat flux (SHF), which is related to air temperature and wind speed. These variables are also overestimated at the margins by MAPI-1w (Fig. 9d and e).

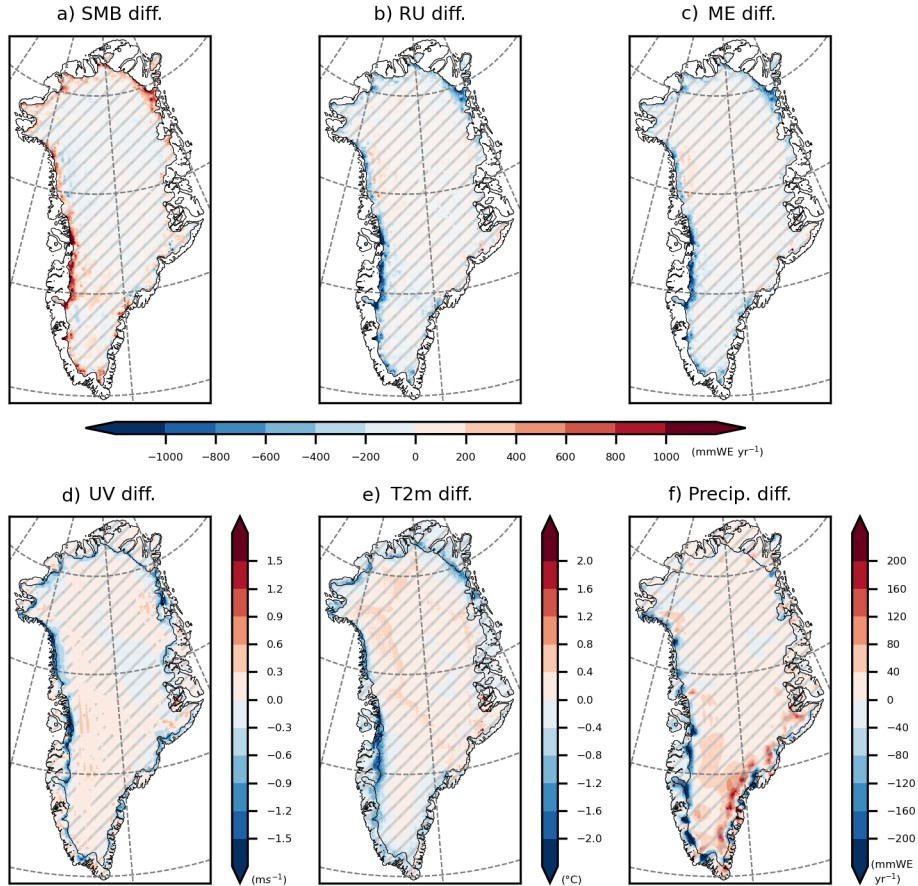

**Figure 9.** Differences (2w minus 1w) of a) SMB (mmWE yr$^{-1}$), b) Runoff (RU, mmWE yr$^{-1}$), c) meltwater production (ME, mmWE yr$^{-1}$), d) 10m-wind speed (UV, ms$^{-1}$), e) 2m-temperature (T2m, °C), and f) total precipitation (Precip. are the sum of rainfall and snowfall) between the MAPI-2w and MAPI-1w simulations for 2171 – 2200. Insignificant differences are hatched (smaller than the interannual variability of 2171-2200).

The underestimation of SMB in MAPI-1w is due to an overestimation of the melt-elevation feedback by the offline correction when interpolating MAPI-1w onto the PISM grid, as compared to the explicit consideration of this feedback in MAPI-2w. This correction is based on the temperature-altitude relationship to account for the melt-elevation feedback, which alters the SMB

and related variables. The correction applies local linear gradients according to altitude differences between the two considered grid (MAR and PISM). We compare here, on the MAR grid, the yearly evolution of the altitude differences between the two experiments (coupled and uncoupled) with the evolution of the temperature differences. This comparison is realised in two specific locations: inside the ice sheet and on the margin (Fig. 10a). This provides an insight into the local SMB gradient as simulated by the fully coupled MAR experiment. We notice that on the margin, differences in altitude between the two MAR grids ($\Delta$SH) explain only 61% (69% for melt) of the changes in temperature differences ($\Delta$T2m and $\Delta$ME respectively), compared to the interior of the ice-sheet, where these relationships are much more dependent, with $R^2$ values of 0.99 and 0.94, respectively. In our example (Fig. 10a), the modifications of topography in the 2-way coupling experiment have modified this linear relationship with the temperature from -0.4 °C/100m inside to -0.1 °C/100m. Similar relationships are illustrated for melt differences (Fig. 10b), confirming the modification in the linear dependence with changes in surface elevation. To further investigate this, we will compare these gradients, obtained by comparing the MAR simulations with and without changes in topography over time, with the gradients used by the offline correction. These gradients are calculated locally, taking into account the differences in altitude and in the variable considered with the surrounding grid cells. For example, for temperature, we find gradients of -0.69 and -0.65 °C/100m in 2200 respectively for the same locations as in Fig. 10, inside the ice sheet and on the margins. Although these gradients have different absolute values compared to those obtained by comparing the two MAR simulations over time, the difference between the two regions is smaller. The temperature gradient applied to the margin of the ice sheet follows a similar dependency to the altitude that the gradient in the interior of the ice sheet. This explains the exaggeration of temperature and temperature-dependent variables (melt, SMB, etc.) on the margins by the correction. The correction uses a gradient which is too large and does not represent the processes leading to the mitigation of the temperature-altitude dependence, and, consequently, the melt-elevation feedback. Finally, we can note that beyond a 350 m drop in altitude, the association between changes in altitude and temperature (or melting) at the ice sheet margin exhibits a behaviour similar to the relationship observed within the ice sheet. Further confirmation is required through experiments involving larger elevation variations that exceed those explored in our current simulation. All these comparisons highlight two main findings: (1) the linear-offline correction of SMB is no longer valid at the ice sheet margins; (2) the changes in the linear relationship between temperature and altitude driving the melt-elevation feedback lead to mitigation of this feedback along the ice sheet margins.

The mitigation of the melt-elevation feedback in the MAR-coupled simulation can be explained by the alteration of local wind and temperature patterns near the margins of the GrIS. The evolution of the topography in the coupled simulation (for instance, Fig. 11e) leads to a reduction in the melt increase with lowering elevation. In general, the production of meltwater is the result of a positive energy balance at the surface. In our study, we focus on sensible heat fluxes (SHF), which are taken into account in this energy balance. SHF is important as it depend on wind speed and temperature, both of which are influenced by alterations in topography. Therefore, we investigate here differences in these two parameters between MAPI-2w and -1w (Fig. 11b and d).

The near-surface temperature, as well as the north-south wind component, are altered along the margin, specifically in the west part of the GrIS in the fully-coupled simulation. To illustrate this, we compare the vertical temperature and wind speed patterns above both simulation topographies along a transect crossing the ice sheet. The example illustrated in Fig. 11

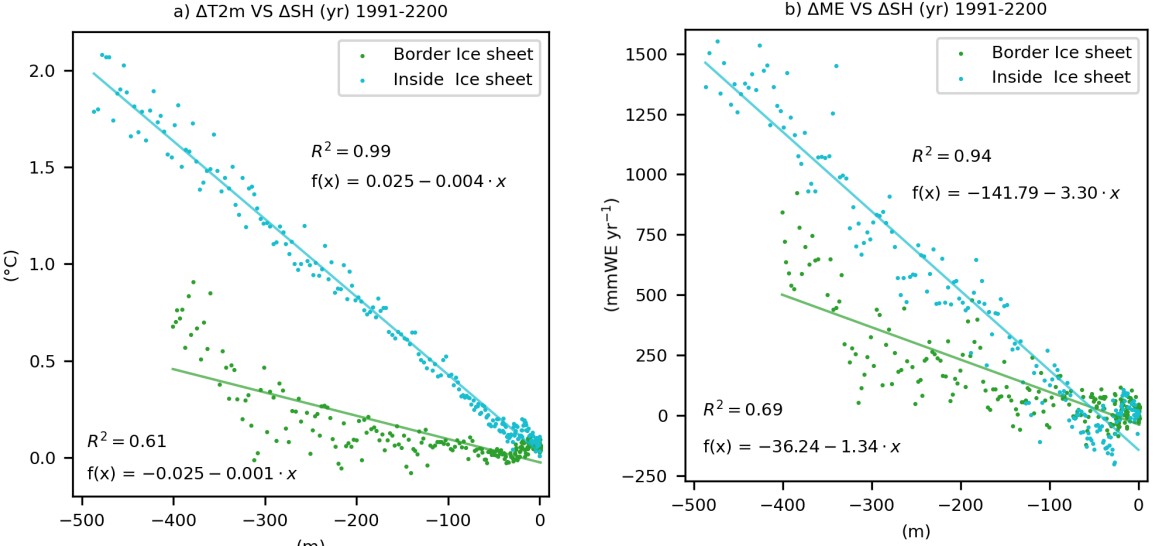

**Figure 10.** Association of the yearly (1991 – 2200) differences on MAR grid in surface elevation ($\Delta$SH, m) with a) differences in 2m-temperature ($\Delta$T2m, °C) and b) meltwater production ($\Delta$ME, mmWE yr$^{-1}$) between coupled (MAPI-2w) and uncoupled (MAPI-1w) simulation for a MAR grid cell (25 km) inside the ice sheet (49.26 ° W, 67.05 ° N, in blue) and one at the boundary with the tundra (48.83 ° W, 67.05 ° N in green). These grid cells are located on the same section of the West Greenland than in Fig. 11. Regressions are presented in the respective colours.

highlights that the north-south wind component (v-wind, positive northward) is larger in the uncoupled simulation on the grid
cell on the ice sheet margin (inside the ice sheet, Fig. 11c and d). Furthermore, the mean near-surface temperature that appears on the 2200-topography (coupled MAR) on the same grid cell of the ice sheet margin is lower than the temperature computed on the uncoupled MAR topography while at a lower altitude (Fig. 11a and b). The changes in wind and temperature in the MAR-coupled simulation provide an explanation for the reduced melt increase with lowering elevation. Note that general wind speed, as well as west-east wind component differences, are presented in the Supplement (Fig. S7).
The temperature changes and the decrease in (v)-wind speed observed in the coupled MAR simulation, in comparison to the uncoupled one, could be explained by local modifications in wind regime and of the margin morphology. These modifications have a mitigating effect on surface melt. In general, as depicted in Fig. 11a, the uncoupled simulation exhibits a greater presence of warm air at the periphery of the ice sheet, where the original topography acts as a barrier preventing deep air intrusion. In the coupled simulation, warm air can penetrate further inland and dissipate, resulting in only modest increase in melting. On
the other hand, in actual conditions, barrier winds occur when air masses from the tundra cross the ice sheet, which acts as an orographic barrier (Van den Broeke and Gallée, 1996). These winds induce warm air advection from the tundra, which is warmer than the surrounding air over the ice sheet. This warm air, confined by the orographic barrier, results in northward

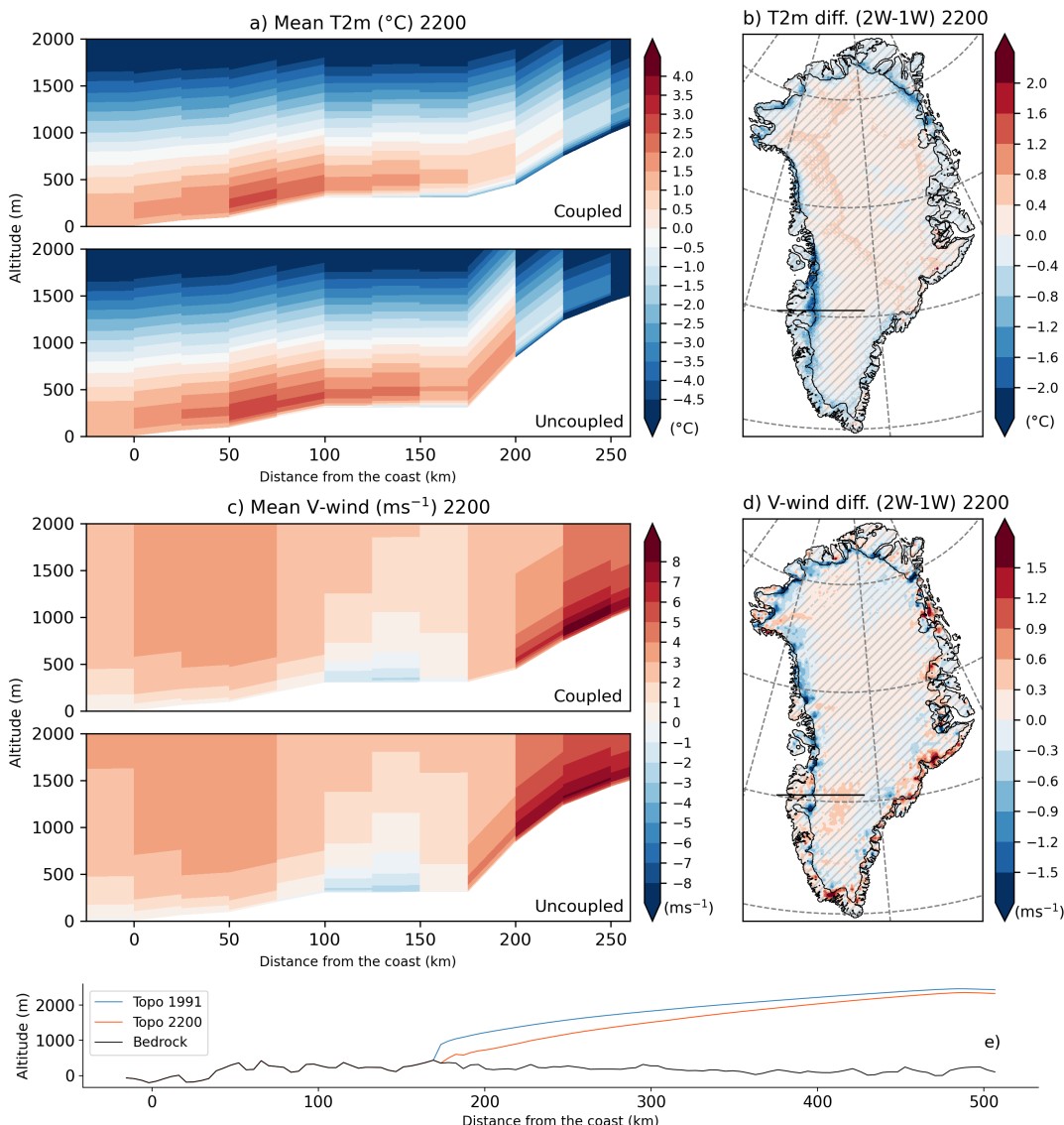

**Figure 11.** Cross sections along 66.64–67.35 ° N of a) temperature (T2m, °C) and c) V-wind component (north-south, positive northward, ms$^{-1}$) on the MAR grid for the coupled (above) and the uncoupled (below) simulation. Differences in b) 2m-temperature (°C) and d) V-wind component (ms$^{-1}$) between 2-way and 1-way experiments of MAR and PISM (MAPI-2w - MAPI-1w) in 2200. Non-significant differences (lower than the interannual variability over 2171–2200) are hatched. T2m and V-wind are interpolated in PISM grid (b and d) to be consistent with other figures, T2m is corrected with the offline correction but not the V-wind as it is not related to surface elevation. e) Cross section along black lines in b) and d) on the west coast of the PISM grid (similar to the cross-section in a) and c) subplots) on MAR grid 66.64–67.35 ° N) of the surface elevation (m) of the MAPI-2w in 2200 (in red), in 1991 in blue (fixed topography in MAR for MAPI-1w simulation) and the bedrock in black.

winds along the west coast of the ice sheet, leading to high temperatures and increased wind speeds along the margins of the ice sheet. Consequently, high-melt events occur due to these high temperatures and wind speeds. As the orographic barrier weakens due to the thinning and retreat of the ice sheet (Fig. 11e) in MAPI-2w compared to MAPI-1w, the barrier winds could potentially diminish, as suggested by the decrease in the v-wind component (south-north) in the coupled experiment. This further would result in temperature disparities between the two experiments, as less warm air advection occurs due to the weakened barrier winds in MAPI-2w. This could impact the local gradient of melt/temperature with surface elevation, and then could lead to a mitigation of melt in MAPI-2w compared to MAPI-1w, where the barrier winds remain unnaffected due to unaltered topography.

## 4 Discussion

The feedback between topography and local atmospheric circulation, as highlighted here, introduces additionnal uncertainty into the SLR projections. These projections are already subject to uncertainties related to ice dynamics modelling. For instance, whithin ISMIP6, SLR estimates for 2100 range from 6.5 to 13.5 cm under the same climate forcing conditions (MAR forced by MIROC5 using the RCP8.5 scenario), depending on the choice of ISMs and experiments (Goelzer et al., 2020). ISMIP6 experiments can be compared to our MAPI-1w simulation, as they employ a methodology that uses MAR outputs corrected offline for the melt-elevation feedback. However extending simulations using such a method could introduce uncertainties coming from the evolving topography of the ice sheet and its interaction with near-surface climate, as discussed in this study.

In a related study by Le clec'h et al. (2019), a methodology similar to ours was employed. They used MAR and the GRISLI ISM (Quiquet et al., 2012) to represent GrIS until 2150. The main difference between their study and ours, besides the ISM used, lies in the large-scale forcing field to force MAR (i.e. MIROC5, a CMIP5 model using the RCP8.5 scenario). The future climate projected by MIROC5 is less warm compared to CESM2, with a difference of approximately $+1.5\,°C$ (Hofer et al., 2020). Consequently, the SLR contributions are consequently well higher when using CESM2. For instance, the warming level projected by MIROC5 for 2100 is reached as early as 2080 in the case of CESM2. Regarding their 2-way coupled simulation, given the disparity in the warming scenarios used for the coupling (MIROC5 RCP8.5 vs CESM2 SSP5-8.5), their MB results in 2100 are similar to those of our MAPI-2w experiment in 2080. From a dynamical perspective, both studies observe a similar overall pattern of speedup in the ice interior and slowdown at the margins of the GrIS towards the end of their respective simulations.

In contrast to Le clec'h et al. (2019), who extended their 1-way simulation until 2150 using constant MAR outputs (SMB and ST), we decided to extend our MAPI-1w simulation beyond 2100 by repeating the last 10 years of large-scale forcing from CESM2 to run MAR until 2200. This was done to allow time for the snowpack to stabilise under the influence of the new, warm, stabilised climate conditions from 2101 to 2200. The main distinction between our approach and that of Le clec'h et al. (2019) is that, after 2100, our MAR simulations with a fixed topography continued to run with repeated CESM2 forcing fields. In contrast, in their corresponding simulation, MAR did not continue beyond 2100, and SMB is therefore assumed to be constant as long as the climate is stable. We compared our 1-way and 2-way experiment results with a simulation where the last

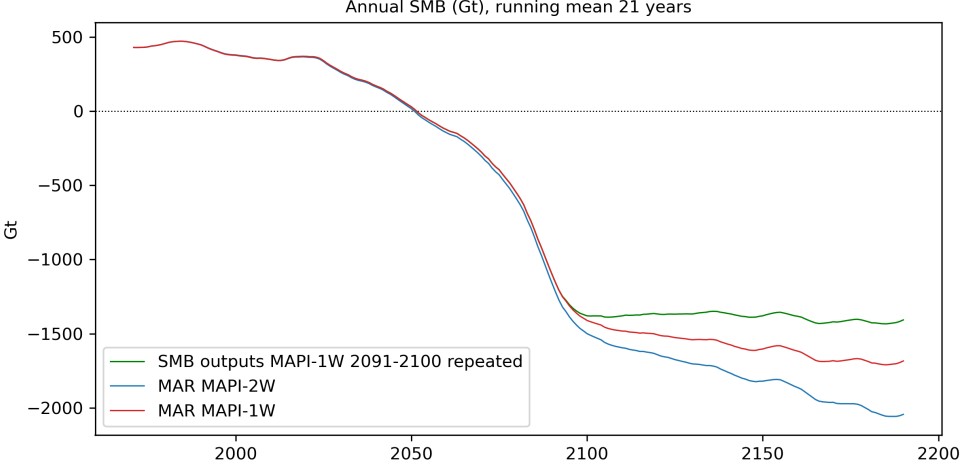

**Figure 12.** Yearly surface mass balance (SMB) integrated over all the ice sheet as simulated by MAR uncoupled (no update of the topography, in red), by MAR coupled (topography updated each year, in blue) and by MAR uncoupled until 2100 and last 10 years of SMB outputs repeated until 2200 (in green).

10 years of MAR outputs from MAPI-1w (the same 10 years as the repeated CESM2 inputs) were used directly to force PISM, extending the MAPI-1w simulation from 2100 to 2200 without running MAR any further. This approach was similar to the one used by Le clec'h et al. (2019) in their 1-way experiment. It appears that in MAPI-1w, SMB continues to decrease for decades compared to the repeated MAR-fixed outputs (Fig. 12). This indicates that, even without additional warming, the ablation
area of the ice sheet continues to expand after 2100. The snowpack turns from an accumulation state into an ablation state over a larger portion of the ice sheet. This transition requires decades to stabilise before reaching a stable meltwater retention capacity. As demonstrated in the RetMIP exercise (Retention Model Intercomparison, Vandecrux et al., 2020), the time required to stabilise meltwater retention capacity is likely model-dependent due to parameters such as maximum liquid water retention within the snowpack and the height of the considered snowpack-layer. This study also emphasises that SMB (through runoff)
cannot be considered stable as soon as warming stops. During the response time of the snowpack, meltwater saturates deep layers, causing it to become warmer and denser, which reduces its capacity to retain meltwater. Once the snowpack reaches its maximum retention capacity, it transforms into impermeable firn or bare ice. Due to the method used in extending the 1-way simulation, this process is not considered in Le clec'h et al. (2019). Therefore, the comparison of our respective methods for representing the melt-elevation feedback (1-way vs. 2-way coupling) lacks a common physical basis.
Another significant aspect of our method under discussion relates to the spatial resolution of the MAR model. To reach a balance between computational efficiency and adequately representing the SMB within the ensemble, we opted for a relatively coarse spatial resolution of 25 km. At the scale of the entire GrIS, this resolution proves to be a viable choice for capturing the global SMB evolution, as supported by previous research (e.g. Fettweis et al., 2020). However, at a finer scale of analysis, this resolution may compromise the accurate representation of local wind and temperature patterns. In some cases, a grid point

might span an area as large as the ablation zone, introducing potential inaccuracies (Van de Wal et al., 2012; Hermann et al., 2018). RCMs remain sensitive to horizontal resolution, particularly when aiming to accurately represent SMB and surface energy balance in specific areas. The local representation of processes and surface topography by the model plays a crucial role in this sensitivity (Franco et al., 2012; van de Berg et al., 2020).

## 5 Conclusions

The coupling of the RCM MAR and the ISM PISM is presented here following the SSP5-8.5 scenario as simulated by CESM2. The 2-way coupling is compared to a 1-way and a 0-way (uncoupled) experiments to evaluate the importance of the melt-elevation feedback.

The first aim was to study what became GrIS in 2200 by applying such extreme conditions. Our fully-coupled simulation projects a contribution of 64 cm to SLR by 2200 with a stabilised climate since 2100 of $+7\,°C$ compared to our reference
period ($1961-1990$). Until 2100 our results are comparable with results obtained in other studies (e.g., Goelzer et al., 2020).

The most effective approach for representing melt-elevation feedback involves fully coupling an atmospheric model with an ice sheet model. Neglecting this feedback leads to an underestimation of the projected sea-level rise contribution by 10.5 %. When comparing two methods to account for the melt-elevation feedback (coupling and offline correction), we highlight that the corrected SMB from the MAR model underestimates the coupled-SMB by 2.5 % when interpolated on the PISM grid using
this correction.

The offline correction is no longer valid on the ice sheet margins because it fails to consider the mitigation of temperature lapse rates, and consequently melt lapse rates, due to changes in topography, such as retreat and slope alterations. These changes influence the wind regime at the margins of the MAR-coupled simulation. The mitigation of melt rates depends on the reduction of sensible heat flux due to changes in local wind regimes and temperature lapse rates. A hypothesis to explain these
local changes around the ice sheet margins involves modifications of barrier wind regimes. These winds typically act along the ice sheet, transporting warmer air from the tundra, enhancing the surface melt, and increasing the northward wind speed along the west side, for instance. The orographic barrier, crucial for the formation of such winds, diminishes with the evolving topography in MAPI-2w. Consequently, enhanced melt could be mitigated, as well as melt-elevation feedback. However, further investigation is required to validate this assumption and to understand the underlying physical processes responsible for
these local wind regime changes initiated by modified surface topography in a warmer climate. A moment budget comparing both simulations (MAPI-1w and -2w) could help identify differences in local atmospheric circulation patterns leading to such variations in representing melt rates at the margins (van Angelen et al., 2011). Additionally, a complete analysis focusing on characteristics such as recurrence and synoptic situations favourable to the development of these wind events will be necessary to gain a better understanding of the physical processes involved. In conclusion, the coupling is essential to update the surface
topography in the RCM and to consider all interactions between the near-surface atmosphere and the new morphology.

By extending our simulations beyond the available period of large-scale forcing of MAR (CESM2), we highlighted that assuming SMB stability when climate conditions become stable is not valid. It is essential to consider the response time of

the snowpack to warming rates and its capacity to retain meltwater. Conducting further sensitivity tests will be necessary to determine if the response time of the ice sheet in stabilising the snowpack depends on the parameterisation of snow layer conditions, such as the maximum liquid water content in a layer or the height of the snowpack simulated by the model.

In conclusion, our study emphasises the significance of topography in influencing the local atmospheric pattern, particularly in shaping local wind regimes along the ice sheet margins, in addition to the well-known melt-elevation feedback. Since these processes influence melt in opposite ways, the melt-elevation feedback is mitigated by the evolving topography. This aspect is not accounted for in the commonly used offline correction, which aims to avoid computationally time-consuming coupling between climate and ice sheet models to perform MB projections and to consider the melt-elevation feedback. Neglecting this negative feedback leads to an overestimation of 2.5 % (equivalent to 1.6 cm in 2200) in the SLR contribution compared to the result obtained with full coupling. Such oversight introduces uncertainty in projections (e.g. ISMIP project) that fail to consider this process, which can be accurately represented through an atmosphere-dynamics coupling.

*Code and data availability.* The MAR code used in this study is tagged as v3.11.3 on https://gitlab.com/Mar-Group/MARv3 (last access: 25 October 2023) (MARTeam, 2023). The PISM code used is tagged as PISMv1.2.2 on https://github.com/pism/pism/releases/tag/v1.2.2 (last access: 25 October 2023). Other coupling scripts are also available upon request by email (alison.delhasse@uliege.be). The main MAPI outputs used in this study are available on Zenodo (https://doi.org/10.5281/zenodo.10066184; Delhasse, 2023). Other specific results are also available upon request by email (alison.delhasse@uliege.be).

*Author contributions.* AD and JB conceived the study. AD and JB performed the simulations. AD led the writing of the manuscript. AD, JB, CK and XF discussed the results. All co-authors revised and contributed to the editing of the manuscript.

*Competing interests.* The authors have the following competing interests: Xavier Fettweis is an editor of The Cryosphere.

*Acknowledgements.* This research has been supported by F.R.S.-FNRS, the Fonds Wetenschappelijk Onderzoek-Vlaanderen (FWO) under the EOS project no. O0100718F and the European Union's Horizon 2020 research and innovation programme under the PROTECT project no. 869304. Computational resources used to perform MAR-PISM simulations have been provided by the Consortium des Équipements de Calcul Intensif (CÉCI), funded by the F.R.S.FNRS under grant 2.5020.11.

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
