# Peer review of "Coupling the regional climate MAR model with the ice sheet model PISM mitigates the melt-elevation positive feedback"

_The Cryosphere, 2023_

## Referee Comment (RC1)

Review of "Coupling the regional climate MAR model with the ice sheet model PISM mitigates the melt-elevation positive feedback" by Alison Delhasse et al.

The authors present an original simulation framework which involves a 2-way coupling on yearly basis between a regional climate model (MAR) forced by a global climate model (CESM2), and an ice sheet model (PISM), to simulate the evolution of the Greenland ice sheet between 1990 until 2100. The simulations are extended to 2200 by randomly sampling the last 10 years of CESM2. The modelled surface mass balance by MAR, and the modelled surface lowering by PISM using the 2-way coupling are then investigated. Finally, the melt evaluation feedback is quantified by comparing the 2-way coupling experiment to two simplified coupling experiments that can be applied on existing MAR simulations on a fixed topography.

This study contains an innovative simulation that for the first time combines different models widely used in the research community (MAR, PISM, CESM), thereby pushing the boundary and providing new insights on ice sheet/atmosphere interactions on decennial time scales. The methods are robust and suitable to model fast (within 100 years) deglaciation in a more realistic manner than previously used. Some parts of the methods could be explained in a simpler manner (the initialisation), while adding details in some other parts would improve readability and reproducibility (the offline correction). The significance of this work is high, in the way that it is one of the few studies to explicitly resolve the 2-way interaction between ice sheet and climate without the use of heavily parameterized equations for the surface mass balance used in ice sheet models. The figures including the supplementary material are clear, and the paper is well written for the most part.

Therefore I recommend publication in The Cryosphere, and I would like to congratulate the authors on implementing such a coupling. The following part contains some suggestions that could be implemented to further improve/clarify the manuscript.

**Major comments**

- The algorithm used to correct for elevation differences (the offline correction) is a key part of the methodology, and it is extensively mentioned in the results and discussion. I would recommend to provide a more detailed description of this correction in the methods section, and possibly mention how this algorithm was optimised for the Greenland ice sheet. This would help the reader to better understand the main conclusion of this manuscript. For instance, it is not clear now if the surface temperature was corrected with a lapse rate, and if so, using which value ?

- 25km horizontal resolution seems rather coarse to represent the narrow ablation zone in the most part of the Greenland ice sheet. The observed SMB can vary by a factor 2 within such a distance (e.g. on the K-transect, Van de Wal et al 2012), or even contain the entire ablation zone (e.g. on the Q-transect, Hermann et al 2018). While I acknowledge that the aim of this study is not to accurately model the SMB, it is likely that such a relatively coarse resolution strongly deteriorates the modelled wind and temperature patterns near the edges, therefore significantly changing the turbulent heat fluxes, and therefore surface melt. I would recommend to mention this in the discussion, and possibly refer to some studies which have shown that RCMs are still not yet accurately modelling turbulent heat fluxes near the edges of the ice sheet (e.g Fausto et al. 2016), or are sensitive the horizontal resolution (e.g. Franco

2012, van de Berg et al. 2020)

- One of the main results of the research is found in p12 L12 "ME depends on the sensible heat flux (SHF) related to air temperature and wind speed which are also overestimated on the margins by MAPI-1w (Fig. 7d and e)". I would recommend mentioning this very interesting result in both the abstract and the conclusion, since this is a key mechanism explaining the lower lapse rates in the coupled experiment.

- In the discussion, a very interesting link is suggested between the lower melt rates in the 2W coupled experiment and the mitigation of barrier winds due to drastic surface lowering. While this is a plausible explanation for the changes, this would require further analysis to properly quantify. For instance the increase in surface slope is also expected to affect the katabatic forcing in the momentum budget. Therefore I would recommend to either perform a more detailed analysis of modelled wind patterns, e.g. by investigating the entire momentum budget (van Angelen et al 2016) between the 1W and 2W experiments, or to mention in the text that changes in barrier winds are just one (of the possibly many) possible effects of surface lowering. In the conclusion (p18 L5) the reduction in barrier winds is now stated as a fact yet it has not been demonstrated in this study.

**Minor comments**

- p2 L25 The statement that the offline correction works well as long as SMB is mainly dominated by elevation could be reformulated. In principle there is no reason to believe that the SMB is a linear function of elevation, yet this is what is observed in the field.

- p2 L34 "What becomes GrIS" should be reformulated

- p3 L16 "good performance". Providing some numbers would be useful to better describe the uncertainties in modelled SMB by regional climate models. It would also help to mention that the evaluation of MAR by the authors (Delhasse et al, 2020) was using a higher horizontal resolution in MAR (15 km). See also Major comment #2.

- p6 It would be useful to extend Figure 1 with the initialisation steps to better understand section 2.3.1.

- It is not clear why the surface mass balance is sometimes referred to as "(surface) mass balance" (e.g. p7 L29) or "surface mass balance". Using the same would improve readability.

- p8 I believe there is something wrong with the notation of mass loss in L2 :"-50 Gt.10−3". Should it be 50 10.3 Gt ? Also in L4.

- p1 L5 What is meant exactly by "as well" ? Do the authors refer to the performance of degree-day models in ice sheet models ? Please be more specific.

- p8 The unit of the y-ax in fig 8b is missing. Adding the variables of each ax would also increase readability.

- p14 L3. Why is only the north-south wind component investigated, and not the wind vector or even the entire vertical profile of modelled wind speeds ? The latter would give a clear indication of changing boundary layer structure and therefore surface fluxes.

**Technical comments**

- o1 L10 "to" − > "for"

- p1 L12 "avoid"

- p5 L8: (Franco et al, 2012)

- p5 "2.3.1 Inisialisation"

- p17 L16 "Do"

**References**

Van De Wal RSW, Boot W, Smeets CJPP, et al (2012) Twenty-one years of mass balance observations along the K-transect, West Greenland. Earth Syst Sci Data 4:31–35. https://doi.org/10.5194/essd-4-31-2012

Van de Berg WJ, van Meijgaard E, van Ulft L (2020) The added value of high resolution in estimating the surface mass balance in southern Greenland. Cryosph 1–28. https://doi.org/10.5194/tc-2019-256

Franco B, Fettweis X, Lang C, Erpicum M (2012) Impact of spatial resolution on the modelling of the Greenland ice sheet surface mass balance between 1990-2010, using the regional climate model MAR. Cryosphere 6:695–711. https://doi.org/10.5194/tc-6-695-2012

Van Angelen JH, van den Broeke MR, van de Berg WJ (2011) Momentum budget of the atmospheric boundary layer over the Greenland ice sheet and its surrounding seas. J Geophys Res 116:. https://doi.org/10.1029/2010jd015485

Delhasse A, Kittel C, Amory C, et al (2020) Brief communication: Evaluation of the near-surface climate in ERA5 over the Greenland Ice Sheet. Cryosphere 14:957–965. https://doi.org/10.5194/tc-14-957-2020

Fausto RS, van As D, Box JE, et al (2016) The implication of nonradiative energy fluxes dominating Greenland ice sheet exceptional ablation area surface melt in 2012. Geophys Res Lett 43:2649–2658. https://doi.org/10.1002/2016GL067720

Hermann M, Box JE, Fausto RS, et al (2018) Application of PROMICE Q-Transect in situ accumulation and ablation measurements (2000-2017) to constrain mass balance at the southern tip of the Greenland ice sheet. J Geophys Res Earth Surf 123:1235–1256. https://doi.org/10.1029/2017JF004408

---

## Author Response (AR1)

Dear Alison Delhasse and co-authors,

I want to truly thank you and your co-authors for the sincere and thorough answers to many of the review comments.

Both reviewer acknowledge the significance of your study as it provides new insight on ice-climate interaction. They both appreciate your discussion on the barrier winds as an explanation for differences in centennial ice evolution.

Concerning your point-by-point answers, I appreciate that you plan to extend the description of the ice-flow model and the offline correction, which will make your manuscript more easy to follow. Moreover, you put a valuable effort into scrutinising differences in lapse-rates between the ice-sheet interior and the margin. However, you stay evasive w.r.t. a more comprehensive analysis of changes in the overall wind pattern. Moreover, I am not sure if my initial comment (access review) on the consequences of the initialisation technique (5x period 1961-1990) is answered. Please reconsider.

On the basis of your point-by-point answers, I invite you to submit a revised manuscript, which will certainly be an improvement. For an objective evaluation of your answers and the implemented changes, I suggest that your revised article enters another review round.

Best,

Johannes Fürst

We would like to thank the editor for his diligent monitoring and constructive comments. Regarding his last comments, we intend to address two of them within this document. Subsequent to providing these responses, we have included a completed version of the point-by-point answers to both reviews with the corresponding modifications in the text.

1/ However, you stay evasive w.r.t. a more comprehensive analysis of changes in the overall wind pattern.

With the objective to be clearer in our manuscript about the general comprehension of the changes in local wind patterns, we improve our paragraph and add, as asked by the first reviewer, general wind speed and west-east wind component analyses (similar figure to Fig. 9 in the original manuscript). Modifications are as follows:

[revised manuscript text omitted]

2/ Moreover, I am not sure if my initial comment (access review) on the consequences of the initialisation technique (5x period 1961-1990) is answered. Please reconsider.

We complete our original answer to the question :

Editor comment: "*If I understand the experimental setup correctly, you introduce an 'initialisation of the coupling' to synchronise PISM and MAR in the period 1961-1990 (P6L3-12). Yet the strategy you suggest implies that the ice-sheet experiences very similar conditions in this 30yr-period for 5 times, meaning an evolution of altogether 150 years. Is that right? If not, ignore the rest. Otherwise, I think that PISM has already seen MAR climate before during the paleo spin-up. Admittedly, MAR has seen the observed ice-sheet geometry, yet with an elevation correction. I understand that you have to move onwards to the modelled geometry at some point in time. In 1991, you furthermore jump from ERA5 to CMIP6 climatic forcing. In this way, you introduce an SMB jump in 1961 and 1991. Moreover, you impose 150 years of additional ice-sheet evolution. I wonder if both is necessary or if you can simply switch both between observed and modelled geometries as well as from ERA to CMIP6 forcing at the same moment in time. If you fear a too large SMB bias for the latter, you could use an anomaly mode in the CMIP6 forcing. I do not see this point too*

*critical as you do not aim for best sea-level projections in this study. Your main conclusions here will not be much affected. Anyway, these choices need dedicated discussion.*"

Thanks for your comment, our section is probably unclear as it stands. We use the 30-year climatology as a stable state assumption and ran actually 5 glacial paleo spinups (5 x 125 k years). Thus, we assume that the GrIS was stable around 1961-1990 and that the GrIS had a similar state 125 ka years ago. So we do not apply 5 x 1961-1990 MAR climatology to PISM, but we apply 5 times paleo conditions to PISM through a complete initialisation of the model thereby using the adapted 1961-1990 MAR climatology as a baseline and adding the temperature anomalies of the paleo time scale, 125 ka. It is true that we do not change the SMB over the paleo spinup but only change surface temperatures. These temperature anomalies propagate through the ice sheet during one glacial cycle modulating the ice rheology, its ability to deform and therefore the ice flow. the SMB field and the change of dynamics determine the new present-day ice sheet, with a new topography. This topography differs from our initial one and therefore would lead to a different SMB field for the same CESM climatology in MAR. Therefore, we run the paleo spin up again. Each time we corrected the stable state topography and climatology with MAR and used the new topography in PISM. We can add that we do not jump from ERA5 to CESM2 in 1991 for MAR, we always run MAR with CESM2 as large-scale forcing, with the historical run until 2014, and ssp5-8.5 from 2014 to 2100.

We first would like to thank the Reviewer#1 for the thoughtful comments which will help to improve our manuscript.

**Major comments**

1. The algorithm used to correct for elevation differences (the offline correction) is a key part of the methodology, and it is extensively mentioned in the results and discussion. I would recommend to provide a more detailed description of this correction in the methods section, and possibly mention how this algorithm was optimised for the Greenland ice sheet. This would help the reader to better understand the main conclusion of this manuscript. For instance, it is not clear now if the surface temperature was corrected with a lapse rate, and if so, using which value ?

It seems that our current description of the offline correction it's not clear and deep enough as the two reviewers pointed out. To address this issue, we propose to add a scheme to illustrate better how this correction is set up, as suggested by Reviewer#2. Furthermore, we propose to revise the explanation as follows:

P5. L. 3-10: "Before any data exchange between the models, data has to be interpolated on the destination grid because the two models were run at two different spatial resolutions (25 vs 4.5 km). The surface elevation simulated by PISM is then interpolated using a four-nearest-neighbour distance-weighted method on the MAR grid at 25 km. For the MAR variables, they are interpolated using the same method on the PISM grid at 4.5 km. However, they are further corrected by considering the difference in altitude between the two grids at the time of interpolation thanks to local vertical SMB/ST gradients. This method is described in (Franco et al., 2012) and is called offline correction hereafter. This method corrects the altitude-dependent variables (such as SMB and ST) by applying a local linear gradient of the variable according to the surface elevation differences between the current MAR grid cell, and the surrounding MAR grid cells (9 grid cells considered here to compute the vertical gradient)."

Become:

"Before any data exchange between the models, data has to be interpolated on the destination grid because the two models were run at two different spatial resolutions (25 vs 4.5 km). The surface elevation simulated by PISM is then interpolated using a four-nearest-neighbour distance-weighted method on the MAR grid at 25 km. The MAR variables are interpolated using the same method on the PISM grid at 4.5 km. However, they are further corrected by considering the difference in altitude between the two grids at the time of interpolation thanks to local vertical SMB/ST gradients. This method is described in (Franco et al., 2012) and is called offline correction hereafter. Firstly, a linear and elevation-dependent gradient (Figure R1) is calculated over the MAR grid by considering the values of the considered variable (SMB at 4.5 km in our example, Figure 1) of the eight surrounding grid cells of the current one. This gradient is specific to each PISM grid cell and is locally determined. An example of this gradient can be found in Figure RS1 in The Supplement. Subsequently, These gradients

are utilised to correct the variable when it is interpolated onto the PISM grid. The correction is performed by multiplying the interpolated variable by the difference in surface elevation between the grid cells in MAR and in PISM. This offline correction is specifically employed to correct variables that are influenced by temperature lapse rate with altitude, namely temperature and derived variables."

[Figure]

**Figure R1.** Steps of the offline correction as described in Franco et al. (2013). After interpolation of a variable (SMB, surface mass balance, in this figure) from a low to higher resolution grid, this variable is corrected to consider the influence of the temperature lapse rate with altitude. The correction is based on a local gradient (d) calculated by considering SMB differences ($\Delta$SMB) between 9 nearest grid cells in the neighbourhood of the current one in the source grid in function of the surface elevation difference ($\Delta$SH). Modified from Wyard (2015).

[Figure]

**Figure RS1.** Surface mass balance (SMB) gradients used to correct SMB as modelled by MAR (25 km) when interpolated on PISM grid (4.5 km) in 2200 by the MAPI-1w run (MAR-PISM uncoupled). Gradients (mm.yr-1/m) are multiplied by the difference in surface elevation to correct the rough SMB.

2. 25km horizontal resolution seems rather coarse to represent the narrow ablation zone in the most part of the Greenland ice sheet. The observed SMB can vary by a factor 2 within such a distance (e.g. on the K-transect, Van de Wal et al 2012), or even contain the entire ablation zone (e.g. on the Q-transect, Hermann et al 2018). While I acknowledge that the aim of this study is not to accurately model the SMB, it is likely that such a relatively coarse resolution strongly deteriorates the modelled wind and temperature patterns near the edges, therefore significantly changing the turbulent heat fluxes, and therefore surface melt. I would recommend to mention this in the discussion, and possibly refer to some studies which have shown that RCMs are still not yet accurately modelling turbulent heat fluxes near the edges of the ice sheet (e.g Fausto et al. 2016), or are sensitive the horizontal resolution (e.g. Franco 2012, van de Berg et al. 2020).

Thanks for your comment, it's an excellent remark that we will add to our discussion, as suggested. The reason why such "coarse spatial resolution" (25km) is used is that it's a compromise to well represent the SMB for all the ice sheet, and the MAR-computation time. Especially in a coupled mode, MAR computation time is very large. To optimize experiment time, we then used the optimal 25 km of horizontal resolution, knowing that SMB over all the ice sheet is not significantly improved by using a finer resolution.

We suggest to add  these explanations at the end of the discussion:

"Another significant aspect of our method under discussion relates to the spatial resolution of the MAR model. To reach a balance between computational efficiency and adequately representing the SMB within the ensemble, we opted for a relatively coarse spatial resolution of 25 km. At the scale of the entire GrIS, this resolution proves to be a viable choice for capturing the global SMB evolution, as supported by previous research (Fettweis et al., 2020). However, at a finer scale of analysis, this resolution may compromise the accurate representation of local wind and temperature patterns. In some cases, a grid point might span an area as large as the ablation zone, introducing potential inaccuracies (Van de Wal et al., 2012; Hermann et al., 2018). RCMs remain sensitive to horizontal resolution, particularly when aiming to accurately represent SMB and Surface Energy Balance (SEB) in specific areas. The local representation of processes and surface topography by the model plays a crucial role in this sensitivity (Franco et al., 2012; van de Berg et al., 2020)."

3. One of the main results of the research is found in p12 L12 "ME depends on the sensible heat flux (SHF) related to air temperature and wind speed which are also overestimated on the margins by MAPI-1w (Fig. 7d and e)". I would recommend mentioning this very interesting result in both the abstract and the conclusion, since this is a key mechanism explaining the lower lapse rates in the coupled experiment.

As recommended, we will add this conclusion in both the abstract and conclusion. As discussed in the next comment, this could be followed by the assumption of the wind barrier that explains these differences in fluxes.

4. In the discussion, a very interesting link is suggested between the lower melt rates in the 2W coupled experiment and the mitigation of barrier winds due to drastic surface lowering. While this is a plausible explanation for the changes, this would require further analysis to properly quantify. For instance the increase in surface slope is also expected to affect the katabatic forcing in the momentum budget. Therefore I would recommend to either perform a more detailed analysis of modelled wind patterns, e.g. by investigating the entire momentum budget (van Angelen et al 2016) between the 1W and 2W experiments, or to mention in the text that changes in barrier winds are just one (of the possibly many) possible effects of surface lowering. In the conclusion (p18 L5) the reduction in barrier winds is now stated as a fact yet it has not been demonstrated in this study.

We agree that to confirm this assumption (changes in barrier wind responsible for changes in melt rate) it should be better to realise a complete wind budget. If it should be really relevant, it is also a bite outside the main goal of the paper, which is to present the coupling between MAR and PISM and an explanation of why both methods of representing the melt-elevation feedback give different results. As this comparison highlighted a new feedback link with the wind regime and which mitigates the melt-elevation feedback, it was important to try to explain it. A wind budget requires also more data than we have, meaning that we need to run again the model to obtain more detailed output concerning wind components, and at different levels. Given all these parameters, we prefer to keep this explanation with barrier wind as the main assumption to explain why we have less melt at the margins in the fully-coupled MAR

version. A wind budget should be an excellent exercise to propose as a perspective of this work and will be of course explained in the discussion.

We could also add that the horizontal resolution, as discussed in the former comment is probably not the best one to realize this exercise. A finer resolution should more suitable to represent this kind of phenomenon, as well as flux budget.

The discussion, conclusion and abstract will be adapted following this comment and the answer.

Following comments 3) and 4), we suggest correcting the conclusion as follows:

[revised manuscript text omitted]

by elevation could be reformulated. In principle there is no reason to believe that the SMB is

a linear function of elevation, yet this is what is observed in the field.

To clarify this sentence we propose to add that SMB is linked to the surface elevation through the temperature lapse rate.

"Using an offline correction works well as long as SMB (particularly melt) is mainly influenced by the surface elevation."

Will be replaced by:

"Using an offline correction works well as long as SMB (particularly melt) is mainly influenced by the surface elevation **through the temperature lapse rate**."

2. p2 L34 "What becomes GrIS" should be reformulated

"First, the aim is (1) to analyse what becomes GrIS in 2200 with this new coupling following an extreme scenario."

will be replaced by:

"First, the aim is (1) to analyse **the evolution of the GrIS by 2200** with this new coupling following the influence of an extreme scenario."

3. p3 L16 "good performance". Providing some numbers would be useful to better describe the uncertainties in modelled SMB by regional climate models. It would also help to mention that the evaluation of MAR by the authors (Delhasse et al, 2020) was using a higher horizontal resolution in MAR (15 km). See also Major comment #2.

In Delhasse et al. (2020) SMB from MAR is not directly compared to SMB-observation, but near-surface climate variables, determinating SMB in MAR are. What we can add here is that MAR forced by reanalyses better perform to represent near-surface temperature that ECMWF-reanalyses themselves compared to in-situ observation. We can also specify that the resolution was finer in the evaluation paper.

4. p6 It would be useful to extend Figure 1 with the initialisation steps to better understand section 2.3.1.

This remark was already formulated by the editor at the first read. We propose so to integrate into Figure 1 the extension realised for the editor's comment:

5. It is not clear why the surface mass balance is sometimes referred to as "(surface) mass balance" (e.g. p7 L29) or "surface mass balance". Using the same would improve readability.

Thanks for this remark, it seems that it was not very clear. We just mean mass balance and surface mass balance. But we will change and clarify with both abbreviations: MB and SMB.

6. p8 I believe there is something wrong with the notation of mass loss in L2 :"-50 Gt.10−3". Should it be 50 10.3 Gt ? Also in L4.

It's not a mistake, but we should specify this is the value of total mass balance, which is negative.

7. p1 L5 What is meant exactly by "as well" ? Do the authors refer to the performance of degree-day models in ice sheet models ? Please be more specific.

To clarify, we propose to change the sentence as follows:

"[…] atmospheric models, which can represent the surface mass balance, usually using a fixed surface elevation, and the ice sheet models, which represent the surface elevation evolution but do not represent the surface mass balance as well as atmospheric models."

Will be rephrased as:

"[…] atmospheric models, which can represent the surface mass balance **(Positive degree day, or regional climate models for instance)**, usually using a fixed surface elevation, and the ice sheet models, which represent the surface elevation evolution. **These last ones do not represent the surface mass balance explicitly as well as atmospheric models.**"

8. p8 The unit of the y-ax in fig 8b is missing. Adding the variables of each ax would also increase readability.

Thanks for these remarks, we will modify them in consequence.

9. p14 L3. Why is only the north-south wind component investigated, and not the wind vector or even the entire vertical profile of modelled wind speeds ? The latter would give a clear indication of changing boundary layer structure and therefore surface fluxes.

In order to demonstrate and illustrate the changes in barrier winds, we have only proposed the north-south component of the wind, which shows that speeds in this direction are less pronounced at the margin of the ice sheet in the coupled mode in MAR. The east-west component, as well as the mean wind speeds, have of course been analysed, but are not illustrated here because the changes observed do not allow us to justify the differences in terms of temperature, and therefore in terms of melting and SMB, between the results from the coupled and uncoupled simulations. These illustrations can of course be added as supplementary material.

**Technical comments**

• o1 L10 "to" − > "for"

• p1 L12 "avoid"

• p5 L8: (Franco et al, 2012)

• p5 "2.3.1 Inisialisation"

• p17 L16 "Do"

Thanks for all these technical comments, they were all considered in the revised manuscript.

**Specific Comments**

In several places, I think the paper would be of better quality if some clarifications or additional explanations were made.
1/ First of all, I think that a more extensive description of the PISM model is required. The description provided in the paper is too technical, if not incomprehensible,for a reader who is not familiar with ice sheet dynamics models. Although a number of references are given in which PISM is described, I think it is important to be able to understand the important points of this section without going to look for the references mentioned. Also, I think it is necessary to define a number of notions, such as SIA, SSA, the Mohr-Coulomb criterion, exponent of the sliding law, flow enhancement factor, etc... This list is obviously not exhaustive and some equations (and their meaning) would allow a better understanding of how the model works. For example, in figure 6 you represent the driving stresses. But how are these driving stresses calculated? Since you mention in the description of PISM the aspects related to the dynamics of the ice, it would be interesting to refer to them more in the analysis of the results, or else to keep in the description of the model only the characteristics used for the analysis of the simulations.

Yes, we agree and explain now a bit more the PISM setting and try to communicate to a broader audience. However, this section is very technical because it should also inform other ice sheet modelers about the ice sheet settings we used, such that they can

reproduce or improve this experiment. Explaining all equations would go beyond the scope of this study and readers should refer to references describing PISM.

We propose to revide the entire section 2.1.2 about PISM description as follows:

"To represent the dynamics and surface elevation of the Greenland Ice Sheet (GrIS), we utilise the Parallel Ice Sheet Model (PISMv1.2.1), a high-resolution numerical ice-sheet/ice-shelf model (Bueler and Brown, 2009; Winkelmann et al., 2011). In PISM, the geometry, temperature, and basal strength of the ice sheet are incorporated into stress balance equations at each time step to determine the ice velocity.

PISM employs two approximations for shallow ice sheets: the Shallow Ice Approximation (SIA) and the Shallow Shelf Approximation (SSA). The SIA is suitable for slowly flowing ice that deforms under its own weight, assuming a strong connection between the ice base and the bedrock. The softness of the ice, affecting its flow velocity, is modulated by an enhancement factor, which we set to $E = 3$ in our experiments. Faster flowing ice, such as ice streams, glaciers, and shelves, is approximated using the SSA. PISM combines both approximations into a hybrid stress balance mode (Bueler and Brown, 2009; Aschwanden et al., 2012).

Basal sliding of the ice over the bedrock introduces basal resistance. The speed of basal sliding is determined by the sliding law, typically a power law relating to the basal shear stress and yield stress. In our study, we adopt the Mohr-Coulomb criterion and use an exponent of $q = 0.6$ for the sliding law.

The model considers basal resistance based on the hypothesis that the ice sheet rests on a till layer. The yield stress represents the strength of this aggregate material at the base of an ice sheet. When yield stress is lower than the driving stress ($\tau d$) there is likely to be sliding, and thus faster velocities can be observed. The driving stress in turn is dependent on the ice thickness (H) and surface gradients (Hs) of the ice: $\tau d \propto Hs$. The thicker and steeper the ice, the higher the driving stress and most probably the ice velocity.

The properties of the till are further approximated by using material properties such as the friction angle. We vary the till friction angle linearly between 5° and 40° with respect to bedrock elevation (between -700m and 700m), following Aschwanden et al. (2016). This variation in friction angle leads to lower friction at lower altitudes and below sea level, resulting in increased surface velocities at the margins of the ice sheet, thus improving the match of flow structure for the glaciers.

To match the present-day extent of the ice sheet, we impose a strong negative surface mass balance (SMB) at the margins of the Greenland present-day ice mask. This setup allows only for ice retreat in our experiments. We also enforce a minimum thickness of 50 m for floating ice at the calving front and utilize the von Mises calving law, which is suitable for glaciers in Greenland (Morlighem et al., 2016). All other parameters are set to default values (University of Alaska Fairbanks, 2019). Our simulations do not consider bedrock deformation or changes in ice-ocean interaction, as we maintain constant submarine melt rates."

2/ The GRIP record does not extend to 125 ka as the signal was perturbed at the bottom of the ice core. While, I think that it shouldn't affect much your results, this should be mentioned. Also, the original publication should be provided (Johnsen et al, 1992) instead of Johnson et al. (2019).

Yes, we cite the dataset here, as it includes several data sources. For clarity, we write now (P4, L24): "The historical time series (Johnson et al., 2019) *includes the temperature* derived from Oxygen Isotope Records from the Greenland Ice Core 25 Project (GRIP, Johnson et al., 1992)"

3/ The initialisation method of PISM and the coupled model could be better explained with a scheme. For example, you mention that during the first step of your initialisation PISM is forced with the 2D temperature anomalies of the last glacial cycle. Are these temperatures correspond to the internal temperatures? If so, these anomalies are not a forcing but an initial state for the next step of the initialisation. Other clarifications are needed: Explain why do you need to equilibrate the vertical temperature profile. This is not clear in the manuscript. How these anomalies are computed? What is the impact of using anomalies instead of absolute values?

For a glacial spinup it is assumed that the state of the ice sheet before a glacial cycle is equal to the one at present day, which means that ice topography and surface temperatures are as well. However, different surface topographies lead to different surface temperatures (which we achieve during our coupled spinup runs.) This is why it is common practice to use temperature anomalies over the last glacial cycle, because the assumption of equal glacial states before and after the glacial cycle only holds when using anomalies. The temperature anomalies are transferred to the surface, which gradually modifies the ice interior temperatures over time. The process of surface temperatures penetrating down into the ice column takes several thousand years due to the thickness of the ice, which can be several kilometers. The internal temperature profile of the ice in turn, determines its softness and deformability, thus affecting the flow velocity of the ice.

To summarize, in our glacial spinup process, we initialize the ice sheet model with temperature anomalies to account for different surface topographies. These anomalies gradually modify the ice interior temperatures over time, influencing the flow behavior of the ice.

We propose to complete section 2.1.3 as follows (since P4, L. 19):

"PISM is forced by yearly ST and SMB from MAR forced by CESM2. To achieve a stable spinup state, we forced PISM with the MAR mean fields (ST and SMB) over 1961 – 1990, when the GrIS was close to balance (Fettweis et al., 2017). However, for a realistic thermodynamics representation of the ice sheet, the temperature evolution of the last glacial cycle has to be considered"

Becomes:

"PISM is forced by yearly ST and SMB from MAR forced by CESM2. To achieve a stable spinup state, we forced PISM with the MAR mean fields (ST and SMB) over 1961 – 1990, when the GrIS was close to balance (Fettweis et al., 2017). However, for a realistic thermodynamics representation of the ice sheet, the temperature evolution of the last glacial cycle has to be considered, ***because the surface temperature slowly propagates down the ice column and determines the vertical ice profile of the ice sheet. The ice profile determines the ice softness and deformability, thus affecting the flow velocity of the ice.***

***For a glacial spinup, it is assumed that the state of the ice sheet before a glacial cycle is equal to the one at present day, which means that ice topography and surface temperatures are as well. However, different surface topographies lead to different surface temperatures (which we achieve during our coupled spinup runs.) This is why it is common practice to use temperature anomalies over the last glacial cycle, because the assumption of equal glacial states before and after the glacial cycle only holds when using  anomalies.***"

We also propose to add Figure R1 in Section 2.3.1 (Initialisation of the coupling) to illustrate both the PISM spinup and the coupling spinup:

[Figure]

**Figure R1**. Steps of the coupling initialisation. Each MAR step corresponds to a 30-year long run over the reference period (1961-1990). And each PISM step consists of a new initialisation cycle of PISM as described in Section 2.1.3.

4/ a) The offline correction method is a key component of the overall paper as is drives the melt-elevation feedback, but the description of the method is very short. Although, I think I understand the basic principles of the method, I found that more details would have been welcome, and possibly a scheme to better illustrate the method.

b) Similarly, I think that the analysis of the results and the comment of Figure 8 are unclear (as well as the conclusions drawn from this figure) are unclear. For example, the authors mention "The dependence is no longer linear" (P12, L20). This just means that the temperature gradients are not the same in the two experiments, if I understood correctly. In other words, the altitude correction is not the same in both experiments. Again, this part of the analysis would be better understood if the authors had given more details about the correction method.

c) Also, I am not sure I agree with the first conclusion of this analysis, namely, "the linear correction is no longer valid in the ice sheet margins". I don't think the analysis leads to this conclusion if the altitude correction is different in the two experiments. Perhaps it would have been better to plot the regressions for each experiment independently (not as anomalies) and to examine the slopes of the regression lines separately.

d) Also, at the ice sheet margins, the behavior is not the same for altitude differences below ~350 m and above. This could be mentioned.

With this comment, we well realised that the description of this offline correction really needs to be revised. We answer these comments in 4 points (a to d) to be clear:

4.a) As asked by the first reviewer too, we propose to revise the description of the offline correction and add an illustration, to better illustrate what's done.

Here are corrections in the text including two more figures (this is the same answer as in review#1, first major comment):

P5. L. 3-10: "Before any data exchange between the models, data has to be interpolated on the destination grid because the two models were run at two different spatial resolutions (25 vs 4.5 km). The surface elevation simulated by PISM is then interpolated using a four-nearest-neighbour distance-weighted method on the MAR grid at 25 km. For the MAR variables, they are interpolated using the same method on the PISM grid at 4.5 km. However, they are further corrected by considering the difference in altitude between the two grids at the time of interpolation thanks to local vertical SMB/ST gradients. This method is described in (Franco et al., 2012) and is called offline correction hereafter. This method corrects the altitude-dependent variables (such as SMB and ST) by applying a local linear gradient of the variable according to the surface elevation differences between the current MAR grid cell, and the surrounding MAR grid cells (9 grid cells considered here to compute the vertical gradient)."

          Become:

"Before any data exchange between the models, data has to be interpolated on the destination grid because the two models were run at two different spatial resolutions (25 vs 4.5 km). The surface elevation simulated by PISM is then interpolated using a four-nearest-neighbour distance-weighted method on the MAR grid at 25 km. The MAR variables are interpolated using the same method on the PISM grid at 4.5 km. However, they are further corrected by considering the difference in altitude between the two grids at the time of interpolation thanks to local vertical SMB/ST gradients. This method is described in (Franco et al., 2012) and is called offline correction hereafter. Firstly, a linear and elevation-dependent gradient (Figure R2) is calculated over the MAR grid by considering the values of the considered variable (SMB at 4.5 km in our example, Figure 1) of the eight surrounding grid cells of the current one. This gradient is specific to each PISM-grid-cell and is locally determined. An example of this gradient can be found in Figure RS1 in The Supplement. Subsequently, These gradients are used to correct the variable when it is interpolated onto the PISM grid. The correction is performed by multiplying the interpolated variable by the difference in surface

elevation between the grid cells in MAR and in PISM. This offline correction is specifically employed to correct variables that are influenced by temperature lapse rate with altitude, namely temperature and derived variables."

[Figure]

**Figure R2.** Steps of the offline correction as described in Franco et al. (2013). After interpolation of a variable (SMB, surface mass balance, in this figure) from a low to higher resolution grid, this variable is corrected to consider the influence of the temperature lapse rate with altitude. The correction is based on a local gradient (d) calculated by considering SMB differences (ΔSMB) between 9 nearest grid cells in the neighbourhood of the current one in the source grid in function of the surface elevation difference (ΔSH). Modified from Wyard (2015).

[Figure]

**Figure RS1.** Surface mass balance (SMB) gradients used to correct SMB as modelled by MAR (25 km) when interpolated on PISM grid (4.5 km) in 2200 by the MAPI-1w run (MAR-PISM uncoupled). Gradients (mm.yr-1/m) are multiplied by the difference in surface elevation to correct the rough SMB.

4.b) Similarly, I think that the analysis of the results and the comment of Figure 8 are unclear (as well as the conclusions drawn from this figure) are unclear. For example, the authors mention "The dependence is no longer linear" (P12, L20). This just means that the temperature gradients are not the same in the two experiments, if I understood correctly. In other words, the altitude correction is not the same in both experiments. Again, this part of the analysis would be better understood if the authors had given more details about the correction method.

Figure 8 highlights the spatial modification of the dependence between altitude and temperature, and consequently between altitude and melt. This is possible because we can compare a simulation where the topography has remained fixed (MAPI-1W) with an identical simulation where only the topography changes (MAPI-2w). So here we compare the results on the MAR grid of these two simulations, before applying the offline correction, each year at the same place. We are not comparing the different gradients used by the correction. Before running the simulations, we expected to find similar dependencies between the evolution over time of topography and temperature over the entire ice sheet. Thus the main expectation was that the gradients found here, by comparing the results over time, would give gradients similar to those calculated by the correction spatially.

Figure 8a.) highlights that in the center of the ice sheet (in blue), when the topography is modified in MAR, melt and temperature evolve linearly with these changes in topography, compared with the results obtained when the altitude remains fixed. In the center we can therefore say that 99% of the temperature changes (94% for melt, figure 8b.) can be explained by the change in surface elevation ($R^2$ = 0.99 and 0.94 rsp.), with a lapse rate of ~ -0.4°C/100m (regression slope) when we compare the annual temperatures obtained with a fixed topography and with a variable topography over time.

However, when we compare the same results for temperature and melt on the inner margins of the ice sheet (relations in green in Figure 8), we notice that only 61% ($R^2$ = 0.61) of the changes in temperature are explained by changes in surface elevation (69% for melt). We all agree that the term non-linearity is perhaps a little strong. But the key message here is that the relationship no longer reflects such a strong dependency ($R^2$ are lower). This proves that other factor(s) explain(s) these temperature changes. We have interpreted this as a factor that mitigates the dependence of temperature on altitude, and therefore the melt-elevation feedback.

The point about the offline correction is that it is based on the theory that changes in surface elevation lead to changes in temperature following temperature lapse rates, and this has a direct influence on melt, SMB and other temperature-dependent variables.

To avoid any confusion in Figure 8 with the correction, we propose to add, in the main text, values of the correction gradients used at these specific locations for both temperature and melt (Table R1). Note firstly that these gradients, for the interior of the ice sheet, are relatively close to those calculated in our example (regression slopes). Secondly, at the margin of the ice sheet, the gradient used in the correction is similar to that inside the ice sheet. However, thanks to our comparison in Figure 8, we know that by modifying the topography, the temperature and melt gradients in the margins are mitigated. It is therefore clear that the correction applied to the simulation with a fixed topography will give higher melt rates than expected at the edges of the ice sheet to the simulation with varying topography where all the processes and feedback resulting from elevation changes are considered. The correction we apply therefore does not take into account the process(es) that attenuate the melt-elevation feedback. This/these process(es) mitigate, as a consequence, the dependence relationship between the temperature and altitude by around 10 to 20%.

The gradients computed for the offline interpolatio from the fixed MAR grid towards PISM grid (MAPI-1w) in the same specific locations as Figure 8 are given in Table R1. Given a 200m difference in surface elevation between the fixed MAR grid and the PISM grid in 2200, this will lead to a correction of 990mm (-4.95 mm x -200m) inside the ice sheet. Similarly, we obtain a correction of 1052.5mm (-4.2 mm x -250m) at the margins. However, following our comparison over time of MAR coupled and running with fixed topography (Figure 8), the gradient in the margin should be much lesser important compared to inside the ice sheet; -4.95 inside VS -4.21 mm/m at the margins as calculated by the correction, and -3.30 VS -1.34 mm/m by comparing the two simulations. So as the gradient does not consider mitigation of the temperature lapse rates, nor the consequent mitigation of the melt-elevation

feedback, the melt correction applied is too important and artificially increases the melt rate. The correction in the margins is too strong and results in underestimating the SMB in these areas.

| Calculated on MAPI-1W | Temperature gradients (°C/m) | Melt gradients (mm/m) |
|---|---|---|
| INSIDE (49.24 °W, 67.07°N) | -0.0069 | -4.95 |
| MARGIN (48.83°W, 67.08°N) | -0.0065 | -4.21 |

**Table R1**. Temperature and mel local gradients as considered by the offline correction in 2200 for the MAPI-1w simulation at two locations, one inside and one on the margin of the ice sheet. These locations are the same as in Figure 8 of the manuscript.

To better explain this figure in the main text, we propose to adapt as following:

P12, L14 - P13, L5:

"The underestimation of SMB in MAPI-1w is due to an overestimation of the melt-elevation feedback by the offline correction when interpolating MAPI-1w towards the PISM grid compared to the explicit consideration of this feedback in MAPI-2w. This correction is based on the linear temperature dependence with the surface elevation to account for the melt-elevation feedback that alters the SMB and related variables. The correction applies local linear gradients according to these altitude differences. We compare, on the MAR grid, the yearly evolution of the altitude differences between the two experiments (coupled and uncoupled) with the evolution of the temperature differences inside the ice sheet and on the margin (Fig. 8a). We notice that on the margin, the dependence is no longer linear (analyse for different other grid cells have been carried out but are not shown here). Inside the ice sheet, the temperature-elevation relation, evaluated as -0.4 °C/100m, remains linear. In our example, modifications of the topography in the 2-way coupling experiment have modified this linear relationship to -0.1 °C/100m of the temperature increase with the surface elevation lowering along the margins. The same relationship is illustrated for melt differences (Fig. 8b), confirming the modification in linear dependence with changes in surface elevation. This highlights two main elements: (1) the linear-offline correction of SMB is no longer valid in the ice sheet margins; (2) the non-linear relationship between temperature and altitude driving the melt-elevation feedback leads to mitigation of this feedback along the ice sheet margins."

Become:

"The underestimation of SMB in MAPI-1w is due to an overestimation of the melt-elevation feedback by the offline correction when interpolating MAPI-1w towards the PISM grid compared to the explicit consideration of this feedback in MAPI-2w. This correction is based on the linear temperature dependence with the surface elevation to account for the melt-elevation feedback that alters the SMB and related variables. The correction applies local linear gradients according to these altitude differences. We compare, on the MAR grid, the yearly evolution of the altitude differences between the two experiments (coupled and

uncoupled) with the evolution of the temperature differences inside the ice sheet and on the margin (Fig. 8a). **We notice that on the margin, differences in altitude between the two MAR-grid (ΔSH) explain only 61% (69% for melt) of the changes in temperature differences (ΔT2m and ΔME respectively), compared with the interior of the ice-sheet where these relationships are much more dependent, with R² of 0.99 and 0.94 respectively.** In our example **(Fig. 8a),** modifications of the topography in the 2-way coupling experiment have modified this linear relationship **with the temperature from -0.4 °C/100m inside** to -0.1 °C/100m. The same relationships are illustrated for melt differences (Fig. 8b), confirming the modification in linear dependence with changes in surface elevation. **We will now compare these gradients, obtained by comparing the MAR simulations with and without changes to the topography over time, with the gradients used by the offline correction. These are calculated locally, i.e. taking into account the differences in altitude and in the variable considered with the surrounding grid cells. For the example of the temperature, we find gradients of -0.69 and -0.65 °C/100m in 2200 respectively for the same locations as in Fig. 8 inside the ice sheet and on the margins. Although in absolute values these gradients are different from those obtained by comparing the two simulations over time, the difference between the two regions is smaller. The gradient applied to the margin of the ice sheet follows a similar dependency to that of the interior of the ice sheet. This explains the exaggeration of temperature and temperature-dependent variables (melt, SMB, etc.) on the margins by the correction, given the use of a gradient that is too large and does not reflect processes leading to the mitigation of the temperature altitude dependence, and consequently, melt-elevation feedback. All these comparisons** highlight two main elements: (1) the linear-offline correction of SMB is no longer valid in the ice sheet margins; (2) **the changes in the linear** relationship between temperature and altitude driving the melt-elevation feedback lead to mitigation of this feedback along the ice sheet margins."

4.c) Also, I am not sure I agree with the first conclusion of this analysis, namely, "the linear correction is no longer valid in the ice sheet margins". I don't think the analysis leads to this conclusion if the altitude correction is different in the two experiments. Perhaps it would have been better to plot the regressions for each experiment independently (not as anomalies) and to examine the slopes of the regression lines separately.

Figure 8 does not present the corrections applied in the two experiments, but the dependencies that exist over time between changes in altitude and temperature, and between altitude and melt, as explained in comment b) above. However, as detailed above, we propose to extend our comment on this figure and add a comparison with actual gradients used by the offline correction.

4.d) Also, at the ice sheet margins, the behavior is not the same for altitude differences below ~350 m and above. This could be mentioned.

That's right, beyond a difference of 300-350m it seems that the relationship is modified. We can point this out in our results. Without reaching values like those inside the ice sheet, it

seems that beyond these differences in altitude, the relationship seems to get closer to the expected one. This could perhaps mean that we are dealing with something temporary, or that after a certain drop in altitude, the well-known dependence between altitude and temperature takes over again in terms of influence. This would have to be verified at different locations in the zones affected by mitigation of melt elevation feedback before any conclusions could be drawn.

5/ The role of barrier winds seems to be a key element to explain the differences in the melt-elevation feedback between MAPI-2W and MAPI-1W. Could it be mentioned more explicitly in the abstract?

As explained in the 1st review, mitigation of barrier winds is one plausible hypothesis to explain why we have here a mitigation of the melt elevation feedback by directly using an evolving topography in MAR. To confirm this hypothesis, it should be better to process a complete wind budget of the two simulations and highlight differences. As it's not really the first aim of this paper, we prefer to keep this as a hypothesis. This is why we prefer to not emphasise this explanation directly in the abstract.

6/ Overall, the paper is written in understandable English. However, I think that the quality of the manuscript would be greatly improved if it were proofread by a native English speaker.

We will make sure to improve the English written expression of the manuscript when we revise it.

**Minor comments**

P1, L3: While MAR is able to diagnose the ice sheet surface mass balance thanks to the implementation of snow/ice layers, this is not the case for many atmospheric models. The statement "atmospheric models which can represent the SMB" should be tempered.

Thanks for your comment, we propose to adjust the sentence as follows:

"This process is called the melt-elevation feedback that can be considered by using two types of models: atmospheric models, which can represent the surface mass balance, usually using a fixed surface elevation, and the ice sheet models, which represent the surface elevation evolution but do not represent the surface mass balance as well as atmospheric models. "
Become

"This process is called the melt-elevation feedback that can be considered by using two types of models: atmospheric models, which can represent the surface mass balance **for some of them, especially polar-oriented regional climate models. But they usually use a fixed surface elevation. And the other side,** ice sheet models which represent the surface elevation evolution but do not represent the surface mass balance as well as atmospheric models. "

P2, L31: "As the coupling is dependent on the used ISM à What do you mean? I guess that you mean that the results of your coupled simulations are model-dependent? Please, clarify.

We specify here that the coupling is model-dependent, specifically concerning the ISM-used, due to the divergence of the results in similar conditions of different models over Greenland (ISMIP Greenland 5 & 6). A coupling is less dependent of the RCM used, as SMB resulting from different RCM are quite similar over the recent period over Greenland. By saying that the coupling is dependent on the used ISM, we refer to and sum up the explanation given in P2, L14-17.

Section 2.13: It seems to be that the abbreviation ST has not been defined before. Also, explain why PISM needs to be forced with ST.

We now explain why we need a forcing of surface temperature. (see answer to major comments above. We introduce "ST" in section 2.1.1.

P4, L-25: The GRIP record was perturbed at the bottom and did not extent to -125 ka. This should affect your results so much but should be mentioned.

Yes, the dataset includes GRIP data but also had other sources so that we could simulate temperatures until -125 ka. See answers to major comments.

P6: L13: MARref forced with CESM2 is also run with PISMsp5 topography (see previous sentence). This sentence is a bit confusing. Please rephrase. Also change PSIMsp5 in PISMsp5.

Yes exactly, both MAR simulations (with CESM2 and ERA5) are running with the same topography from PISMsp5. To be clear, we propose to adjust our sentences as follows:

"The PISMsp5 topography, the last step of the initialisation process, will be the initial state of the different simulations compared here and is used to run the MAR reference simulation over the reference period (MARref). As our projections could not be evaluated, we compared performances of MARref forced with CESM2 to MAR using the PSIMsp5 topography and forced with the observed climate, i.e. the reanalysis ERA5 here (Hersbach et al., 2020). "

Become:

"The PISMsp5 topography, the last step of the initialisation process, will be the initial state of the different simulations compared here and is used to run the MAR reference simulation over the reference period (MARref). As our projections could not be evaluated, **we evaluated the performances of MARref over the present. To do so, we compared MAR results over the current period (1961-1990), with the initialised topography (PISMsp5) forced on one hand, by the ESM used for projections (CESM2) and on the other hand ERA5 reanalysis (Hersbach et al., 2020), considered as observations and well representing current climate.**"

At different places in the manuscript there is confusion between interpolation and aggregation. Outputs coming from a higher resolution model are *aggregated* on a coarser model grid. Variables computed with a lower resolution model are *interpolated* on a finer model grid.

Yes true, we didn't make the distinction, but we will correct that, thanks.

Section 2.4: You should add a comment explaining why you deal sometimes with surface mass balance and sometimes with mass balance. Also, explain (maybe before section 2.4) the difference between both.

The surface mass balance (SMB) is one of the components of the total mass balance of the ice sheet, call here mass balance (MB). The SMB is obtained by using the MAR model and summarises the gains and losses of ice mass at the surface, whereas the total mass balance is the result of computing both SMB and the dynamic of the ice sheet. MB is then the PISM result. In our main text, depending on which component we refer to, we talk about SMB or MB.

To address your request, we propose to specify again which model computes which part of the total mass balance in section 2.2, when we describe the coupling.

P4. L30-31 and P5. L1-2: "The coupling between both models has been performed by exchanging yearly outputs (SMB and ST from MAR, and ice thickness from PISM) on the 1st January of each year for 1991 – 2200 as described in Le clec'h et al. (2019). For MAR, this induces updating the surface elevation and ice extent of the ice sheet at the beginning of each year with PISM results from the previous year, whereas SMB and ST are used as forcing fields for PISM."

"The coupling between both models has been performed by exchanging yearly outputs (SMB and ST from MAR, and ice thickness from PISM) on the 1st January of each year for 1991 – 2200 as described in Le clec'h et al. (2019). For MAR, this induces updating the surface elevation and ice extent of the ice sheet at the beginning of each year with PISM results from the previous year, whereas SMB and ST are used as forcing fields for PISM. **The coupling aims to produce estimations of total MB of the GrIS by simulating dynamical components directly with PISM and using the SMB component as simulated by MAR as forcing for PISM.**"

P8, L4-L5: The sentence is a bit confusing as the climate is not stabilized just before 2100.

We propose to rephrase and complete this part as follows:

"Since there is an acceleration of the mass loss just before 2100, even with a stabilised climate, mass loss is not stabilised in 2200."

Become:

"Since there is an acceleration of **the warming and consequently of the mass loss before 2100, even by stabilising the climate and then the warming after 2100, the mass loss is still increasing until 2200**."

P9, L10: "synoptic features of the large-scale CESM2 forcing" à Could it be illustrated with a figure (in the Supplement part for example)?

Yes, it's a good suggestion. To illustrate the large-scale pattern coming from CESM2 and visible precipitation changes at the end of the 22nd century, we plot these changes with raw CESM2 data (Figure R3). This illustrates the decrease in precipitation on the east coast of Greenland. We also identify an increase in precipitation in the western part. This highlight is added in our result comments in the main manuscript because, mixed with changes in surface elevation explanation, it can explain changes in precipitation observed by 2200 in our coupled results. The text is adapted as follows:

"There is a significant decrease in total precipitation (SF + RF, Fig. 7c) over the southeast due to synoptic features of the large-scale forcing (CESM2, not shown). Conversely, our simulation projects significantly increase over the west and north of Greenland. The increase in the west is a consequence of the ice sheet thinning as clouds can penetrate more inland due to a weaker topographic barrier effect and a delayed condensation due to further lift-up of air masses. **A synoptic pattern coming from the CESM2 forcing is also contributing to this increase in precipitation (not shown).** We attribute changes in snowfall for the north of Greenland to more humidity content associated with atmospheric warming, as this region is particularly dry and cold over present-day conditions."

Unfortunately, as these CESM2-data are incomplete (only available over persistent iced areas) and we do not have access to complete one (CESM2 simulation used as forcing files from MAR in this study), we prefer to not add this in our Supplement.

[Figure]

**Figure R3.** Precipitation (snowfall + rain, mm.yr-1) as simulated by CESM2 over 2071–2100 over the iced surfaces of Greenland region.

P12, L1-2: This sentence is not clear. Je ne comprends pas pourquoi les résultats de MAPI-1W sont contraires à ceux de MAR. I think that the sentence should be rephrased.

As it seems this sentence is confusing, we propose to extend the explanation as follows:

"The MB overestimation by MAPI-1w is contrary to the result of the MAR outputs, where MAPI-2w gave higher melt rates than MAPI-1w (Fig. 4 solid vs dashed lines)."

Becomes:

"The MB overestimation by MAPI-1w could be contrary to the intermediate results from MAR of both MAPI-1w and -2w simulations. **If we look at these results (raw MAR outputs)before interpolation and forcing of PISM, fully-coupled melt rate outputs are higher than in the one-way coupled simulation (Fig. 4 dashed lines). After interpolation, meaning the MAR results interpolated on PISM-grid and which actually forced PISM (Fig. 4 solid lines), MAPI-1w gave higher melt rates than MAPI-2w.**"

P12,L2: Figure 1a indicates that the melt-elevation feedback is taken into account via an offline correction. This seems to be in contradiction with the text.

To be consistent, when we interpolated MAR outputs on the PISM grid in both 1-way and 2-way simulations, we applied the offline correction. The difference is that in the 2-way coupled experiment, as the topography is updated each year in MAR, corrections during this interpolation are negligible compared to corrections applied in 1-way simulation as differences in surface elevation are very short and only due to spatial resolution. The melt-elevation feedback in the full coupling is actually well considered by directly modifying surface elevation in MAR as all SMB components are calculated in changing surface elevation.

P12, L17: To which altitude difference do you refer. This is not clear and justifies a more in-depth explanation of the offline correction method.

We refer here to the difference in altitude between the two grids, MAR and PISM. Concerning the in-deep explanation of the offline correction method, we refer to the answer to the major comment 4.

P13,L6: Remind the link between wind, T2m and SHF, as it appears to be a key mechanism to explain the difference in the melt-elevation feedback in your simulations.

We propose to better explain it as follows:

P13, L6 - P14, L2: "The mitigation of the melt-elevation feedback in the MAR-coupled simulation is explained by the modification of the local atmospheric circulation on the margins around the GrIS. The evolution of the topography in the coupled simulation (for instance, Fig. 9e) causes a decrease in the melt increase with the elevation lowering. As

meltwater production depends on the near-surface temperature and the wind through the sensible heat flux, we compare the vertical temperature and wind speed patterns above both simulation topographies along a transect crossing the ice sheet. The example illustrated in Fig. 9 highlights [...]"

Becomes:

"The mitigation of the melt-elevation feedback in the MAR-coupled simulation is explained by the modification of the local atmospheric circulation on the margins around the GrIS. The evolution of the topography in the coupled simulation (for instance, Fig. 9e) causes a decrease in the melt increase with the elevation lowering. **The production of meltwater is the result of a positive energy balance at the surface. More specifically, changes in sensible heat flux (SHF) account for this which is directly proportional to surface temperature and wind speed. We investigate here differences in these two parameters between the two experiments. They are both directly influenced by changes in surface topography between MAPI-2w and -1w (Fig. 11b and d). The near-surface temperature, as well as the north-south wind component, are altered along the margin, specifically the west part of the GrIS in the fully-coupled simulation (general wind speed, as well as west-east wind component differences, are presented in the Supplement, Fig. S7). To better illustrate that, we compare the vertical temperature and wind speed patterns above both simulation topographies along a transect crossing the ice sheet.** The example illustrated in Fig. 11 highlights [...]"

P13, L6-7: This should be illustrated with a figure.

Actually, this sentence was supposed to summarise the entire following paragraph, and the figure chosen to illustrate all of this is the described one all along this paragraph: Figure 9.

P13, L7-8: Fig9e represents a cross section of the topography and not an evolution of the topography. Please, rephrase. The mitigation of the melt-elevation feedback with elevation lowering is better illustrated with Figs 8a and S4a.

The evolution of surface topography is well illustrated with these cross sections as the fixed topography of MAPI-1w corresponds to the starting topography of the 2way simulation. As we also plot the cross-section of the fully coupled simulation topography in 2200, we consider that we illustrate these changes in surface elevation. Of course, this is also illustrated by Figure 3, with changes all over GrIS, and Fig S4a, which illustrates the differences in surface elevation between the two PISM simulations (from MAPI-1w and -2w). However, in Fig. S4a, differences could be due to, on one hand, the differences in SMB apply to PISM, or, on the other hand, to the model divergence itself. As it could be confusing, we won't add this reference at this place. Furthermore, the goal to illustrate surface changes with the cross-section is to have an idea of the shape modifications too, because the slope is very important when you study wind components.

P14, L4: I don't understand what you mean by "inside the ice sheet" in parenthesis

We just specify here that the grid-cell is in the margin, but still considered inside the ice sheet, so modifications of fluxes and temperature will be important to consider changes in melt.

P17, L16: "Do not consider" à Please rephrase. I suggest something like "We must not consider" or another equivalent formulation.

The sentence would mean that if we miss the consideration of this feedback, this will result in an underestimation of SLR.

We propose to rephrase as follows:

"Missing this feedback will result in underestimating the projected sea level rise contribution of 10.5%"

**Typo and technical comments:**

Sections 2.1.3 and 2.3.1: Inisialisation à Initialisation

P1, L5: Remove "as well as atmospheric models"

P1, L10: corrected to à corrected for

P1, L11: extrapolated à interpolated

P1, L12: avoid à avoids or "prevents from a too expensive coupling"

P2, L26: Remove "as forcing"

P2, L35 "assess the offline method" should be changed in "assess the ability of the offline method…"

P3, L1 "feedback" à feedbacks

P3, L1: Replace "as well as "by " "and which"

P4, L26-27: Provide the resolutions in km, not in meters (as in the other parts of the manuscript).

P5, L5: "For the MAR variables, they are interpolated" à "The MAR variables are interpolated…"

P6: L13: change PSIMsp5 in PISMsp5.

At different places in the manuscript there is confusion between interpolation and aggregation. Outputs coming from a higher resolution model are *aggregated* on a coarser model grid. Variables computed with a lower resolution model are *interpolated* on a finer model grid.

P7, L23: I feel that 10% is not so negligible.

P7, L29: Components refer to SMB. Replace "their components" with "its components"

P8, L4: Replace -200 Gt.10$_{-3}$ by -200 10$_{-3}$ Gt (same thing for L2)

P8, L12: Remove meltwater.

P11, L7: underestimated à underestimates

P12, L3 : become à becomes

P14, L3: at the ice sheet margins / on the ice sheet margins

P18, L14: sensibility à sensitivity

P18, L19: oppositely à in opposite ways ?

**Figures**

Figure 1a : See comment, P12, L2

Fig. 2: The grey band is not visible. Maybe you could choose a darker colour.

Fig 3: Green and red contours cannot be easily distinguished

Fig. 4 caption: (RU, in green) à (RU, in orange). Precise in the figure caption that solid lines correspond to the coupled experiment

Fig.9: Indicate what do the y-axes and x-axes represent (Figs.9a, 9c, 9e). The line thickness of the black lines in Figs.9b and 9d could be slightly increased.

Thanks for all these technical comments, they were all considered in the revised manuscript.

**References**

Wyard, C: Évaluation de la pertinence du couplage MAR-GRISLI sur le Groenland. Mémoire de master en sciences géographiques, orientation climatologie, à finalité approfondie, Liège, Université de Liège, inédit, 96 p. 2015.

---

## Referee Report (RR1)

**Review of the paper "Coupling the regional climate MAR model with the ice sheet model PISM mitigates the melt-elevation positive feedback, by Delhasse et al.**

I would like to congratulate the authors for their efforts to make clearer the manuscript and for their detailed responses to my comments. However, I feel that the paper would be improved if further explanations were provided.

**PISM description:** I acknowledge that the PISM description has been extended. I wonder if the explanations are sufficient for someone not familiar with ice sheet modelling but I understand that the authors ask the readers to refer to the original publications. However, there is a mix between some very specific terms and very general explanations. I give a few examples below:

- Mentioning the value of the exponent(q=0.6) in the sliding law does not make sense if the Mohr-Coulomb criterion is not explained.
- The basis of the von Mises calving law could be explained
- The hypotheses underlying the shallow-ice and shallow-shelf approximations are missing
- Specify that E = 3 is a value often used in most ice-sheet models.

**PISM initialisation:** The explanations do not still sound very clear to me. I suggest to reorganize this section:
1/ Keep the first sentence and continue with "For a realistic thermodynamics representation
2/ Explain why you use anomalies. Note that I do not fully understand the sentence "This is why it is common practice…" Find a clearer explanation or remove this sentence. Also, you should replace "For a glacial spinup, **it is** assumed that" by "For a glacial spinup, **we** assume that…".
3/ Mention at the end of the section that your reference climate is given by the MAR mean fields (ST and SMB) over the 1961-1990 when Greenland was close to balance.

**Offline-correction method:** The explanations of the method are now much clearer. However, I have to admit that I found it hard to understand what the 16 pairs of grid points corresponded to and how they were obtained. I finally came to the conclusions that the following associations are considered: (1,2,4,5), (2,3,5,6), (4,5,7,8) and (5,7,8,9). Is my understanding correct? If so, this should be explained or mentioned in Fig. 1 (or at least in Fig. 1 caption).

Also, Page 6 (L23-24), the fields obtained with the offline-correction method are computed using the eight surrounding grid points, but in Fig. 1 you mention the nine nearest grid points. I think it is a typo error. Otherwise, clarifications should be made.

**MAR initialisation:** It would be interesting to have an idea of how MAR is initialised, particularly with regard to the snowpack model to which the authors refer extensively in the Discussion section.

**Abstract: Line 4:** Positive-degree day models cannot be classified as atmospheric models. They just parameterise the amount of runoff.

**Section 2.3.1:** The velocity fields are compared to those provided by Joughin et al. (2018) over the 1995-2015 period. Differences between modelled and observed velocities are on average ± 80 m s-1 and are much larger on the margins. The authors refer to problems of resolution to explain these differences. However, these differences may also be explained by the fact that Greenland was not in balance in 2015 (and even before). This could be mentioned as an additional possible explanation.

**Supplement:** I guess that the authors did not upload the revised version of the Supplement as there is a mismatch between figure numbering in the main text and in the Supplement.

**Other comments**

I mention below some English mistakes (but the list is not exhaustive). I insist on the need to have the manuscript proof read and corrected by a native speaker. My feeling is that some sections are quite difficult to read with often long sentences which are not always grammatically correct.

P1-L17: Replace "highlighted" by "highlight"

P3-L6: Remove "First"

P3-L21: "input by" → "inputs to"

P3-L25: mention → mentioned

P3-L31: for a doubling of CO2 → for a doubling of atmospheric CO2 concentration.

P6-L3: Add a reference to Section 2.3.1 when you refer to the coupled spinup runs.

P9-L15: of melt-elevation feedback → of **the** melt-elevation feedback (and in other places in the manuscript).

P9-L22: "are only responsible for 10% of the MB" →How is it evaluated ??

P9-L31: "since 1991 of > " → Remove "of" (same remark for P10-L3)

P14-L1: What do you mean with "intermediate results"? Please reformulate

P14-L14: add in MAPI-1w after "from the ME excess"

P14-L15-16: SHF is not plotted in Fig. 9

P15-L7: changes to → changes in

P17-L17: What do you mean with "at depth"?

P19-L4 "They used MAR and GRISLI" → "They used MAR and the GRISLI ice sheet model. Add a reference for GRISLI. For example Quiquet et al. (2012).

**Reference:**
Quiquet, A., Punge, H. J., Ritz, C., Fettweis, X., Gallée, H., Kageyama, M., Krinner, G., Salas y Mélia, D., and Sjolte, J.: Sensitivity of a Greenland ice sheet model to atmospheric forcing fields, The Cryosphere, 6, 999–1018, https://doi.org/10.5194/tc-6-999-2012, 2012.

---

## Author Response (AR2)

Review of the paper "Coupling the regional climate MAR model with the ice
sheet model PISM mitigates the melt-elevation positive feedback, by
Delhasse et al.

I would like to congratulate the authors for their efforts to make clearer the manuscript and for their detailed responses to my comments. However, I feel that the paper would be improved if further explanations were provided.

**PISM description:** I acknowledge that the PISM description has been extended. I wonder if the explanations are sufficient for someone not familiar with ice sheet modelling but I understand that the authors ask the readers to refer to the original publications. However, there is a mix between some very specific terms and very general explanations. I give a few examples below:
- Mentioning the value of the exponent($q=0.6$) in the sliding law does not make sense if the Mohr-Coulomb criterion is not explained.
- The basis of the von Mises calving law could be explained
- The hypotheses underlying the shallow-ice and shallow-shelf approximations are missing
- Specify that $E = 3$ is a value often used in most ice-sheet models.

We have adapted our section to address your comment and the editor's recommendations:

[revised manuscript text omitted]

**Offline-correction method:** The explanations of the method are now much clearer. However, I have to admit that I found it hard to understand what the 16 pairs of grid points corresponded to and how they were obtained. I finally came to the conclusions that the following associations are considered: (1,2,4,5), (2,3,5,6), (4,5,7,8) and (5,7,8,9). Is my understanding correct? If so, this should be explained or mentioned in Fig. 1 (or at least in Fig. 1 caption).

Thanks for your comment, there is a mistake that seems to compromise the correct understanding of this example. There are not 16 pairs of grid-cells compared, but 36! The 9 low-resolution cells selected are compared 2 by 2 in terms of SMB and according to their difference in altitude to obtain a local gradient (the 9 low-resolution pixels closest to the position of the high-resolution pixel) of SMB as a function of differences in surface elevation in the low-resolution grid.

[Figure]

Figure R1. Figure 1 modified.

Figure 1 is corrected in consequences, and we also adapted its caption as follows:

"Figure 1. Steps of the offline correction as described in Franco et al. (2012). After interpolation of a variable (SMB, surface mass balance, in this figure) from a low to higher resolution grid, this variable is corrected to consider the influence of the temperature lapse rate with altitude. The correction is based on a local gradient (d) calculated by considering SMB differences (ΔSMB) between 9 nearest grid cells in the neighbourhood of the current one in the source grid in function of the surface elevation difference (ΔSH). Modified from Wyard (2015)."

Becomes:

"Figure 1. Steps of the offline correction as described in Franco et al. (2012). After interpolation of a variable (SMB, surface mass balance, in this figure) from a low to higher resolution grid, this variable is corrected to consider the influence of the temperature lapse rate with altitude. The correction is based on a local gradient (d) calculated by considering SMB differences (ΔSMB) between 9 nearest low-resolution grid cells in the neighbourhood of the high-resolution grid cell position in function of the surface elevation difference (ΔSH). Modified from Wyard (2015)."

Also, Page 6 (L23-24), the fields obtained with the offline-correction method are computed using the eight surrounding grid points, but in Fig. 1 you mention the nine nearest grid points. I think it is a typo error. Otherwise, clarifications should be made.
It's actually the nine nearest pixels of the low-resolution grid to the position in the high-resolution grid which are considered. We have to change in Page 6, the wording "the eight surrounding grid cells of the current one." which is confusing.

**MAR initialisation:** It would be interesting to have an idea of how MAR is initialised, particularly with regard to the snowpack model to which the authors refer extensively in the Discussion section.

To address this comment we propose to add this short description of MAR initialisation in Section 2.1.1 :
"The polar version of MAR requires a fairly long spinup period to reach an equilibrium state for both the snowpack and the atmosphere. Concerning the snowpack, the parameters that are important for achieving an equilibrium state and representing coherent configuration (temperature, density and liquid water content, Lefebre et al., 2003) are pre-initialised based on former simulations. These simulations have undergone an extensive spinup process spanning over 50 years to establish a coherent representation of the snowpack (Fettweis et al., 2020)."

**Abstract: Line 4:** Positive-degree day models cannot be classified as atmospheric models. They just parameterise the amount of runoff.

We adapt this sentence as follows:
"This process is called the melt-elevation feedback and can be considered by using two types of models: atmospheric models, which can represent the surface mass balance (positive degree day, or polar-oriented regional climate models for instance)."
Becomes:
"This process is called the melt-elevation feedback and can be considered by using two types of models: atmospheric models, which can represent the surface mass balance, or SMB estimates resulting from simpler models such as positive degree day models."

**Section 2.3.1:** The velocity fields are compared to those provided by Joughin et al. (2018) over the 1995-2015 period. Differences between modelled and observed velocities are on average ± 80 m s-1 and are much larger on the margins. The authors refer to problems of resolution to explain these differences. However, these differences may also be explained by

the fact that Greenland was not in balance in 2015 (and even before). This could be mentioned as an additional possible explanation.

Thank you for your comment. We'll add this hypothesis to this section:

"In some fast-flowing glacier regions, differences are well larger. However, the coarse resolution (4.5 km) compared to the proximity of smaller glaciers (500 m), which are solved by the observations, leads to strong deviation in their comparison. **Furthermore, from 1995 to 2015, Greenland was not in balance, and glaciers were already experiencing speed up and retreat (King et al., 2020).**"

**Supplement**: I guess that the authors did not upload the revised version of the Supplement as there is a mismatch between figure numbering in the main text and in the Supplement.
Thank you for your comment. It seems that the supplements have not been loaded for the revised version. We'll be even more careful with the latest version.

**Other comments**
I mention below some English mistakes (but the list is not exhaustive). I insist on the need to have the manuscript proof read and corrected by a native speaker. My feeling is that some sections are quite difficult to read with often long sentences which are not always grammatically correct.

We have revised our manuscript to address this comment and transformed some rather difficult sections to make them more readable. We have also included the following brief comments.

P1-L17: Replace "highlighted" by "highlight"
P3-L6: Remove "First"
P3-L21: "input by" → "inputs to"
P3-L25: mention → mentioned
P3-L31: for a doubling of CO2 → for a doubling of atmospheric CO2 concentration.
P6-L3: Add a reference to Section 2.3.1 when you refer to the coupled spinup runs.
P9-L15: of melt-elevation feedback → of the melt-elevation feedback (and in other places in the manuscript).
P9-L22: "are only responsible for 10% of the MB" →How is it evaluated ??
As presented in Figure S6 in the supplement, these 10% are obtained by comparing differences in SMB as computed on the respective grid of both simulations (PISM from MAPI-1w and -2w) and the same differences computed on the same grid (PISM fully coupled grid). These differences of differences (Fig. S6c) represent about 10% of the real SMB differences obtained (Fig S6a or b). For the sake of clarity, we decided to neglect this 10% because it was not the main cause, and certainly not the physical cause, of the differences between the two simulations.
P9-L31: "since 1991 of > " → Remove "of" (same remark for P10-L3)
P14-L1: What do you mean with "intermediate results"? Please reformulate
"The MB overestimation by MAPI-1w could be contrary to the intermediate results from MAR of both MAPI-1w and -2w simulations."
Becomes:

"The overestimation of MB by MAPI-1w could be contrary to the intermediate results from MAR **before interpolation and correction onto PISM-grid** of both MAPI-1w and -2w simulations."

P14-L14: add in MAPI-1w after "from the ME excess"
P14-L15-16: SHF is not plotted in Fig. 9
P15-L7: changes to → changes in
**P17-L17: What do you mean with "at depth"?**
This sentence has been changed as follows: "In general, as depicted in Fig. 11a, the uncoupled simulation exhibits a greater presence of warm air at the periphery of the ice sheet, where the original topography acts as a barrier preventing deep air intrusion."
P19-L4 "They used MAR and GRISLI" → "They used MAR and the GRISLI ice sheet model. Add a reference for GRISLI. For example Quiquet et al. (2012).

**Reference:**
Quiquet, A., Punge, H. J., Ritz, C., Fettweis, X., Gallée, H., Kageyama, M., Krinner, G., Salas y Mélia, D., and Sjolte, J.: Sensitivity of a Greenland ice sheet model to atmospheric forcing fields, The Cryosphere, 6, 999–1018, https://doi.org/10.5194/tc-6-999-2012, 2012.

---

## Author Response (AR3)

Minor Review of the paper "Coupling the regional climate MAR model with the ice sheet model PISM mitigates the melt-elevation positive feedback", by Delhasse et al.

Additional editor's private note (visible to authors and reviewers only):

LIST OF COMMENTS:

- P4L22: longitudinal stresses —> membrane stresses
- P4L23-35: Remove sentence starting with 'This approximation is suitable for slowly flowing ice …'

Thanks for your 2 comments, we considered these corrections in our manuscript.

- P4L25: Please motivate the choice for E=3 and refer to other studies as requested.
- P4L25-26: I would not speak of the softness of the ice. Use specific terminology (viscosity, fluidity, rate factor)

We adapt our sentences to the two last comments as follows:
"The softness of the ice, affecting its flow velocity, is modulated by an enhancement factor, which we set to E = 3 in our experiments."

is changed to:

"The viscosity of the ice, affecting its flow velocity, is modulated by an enhancement factor E. We set E = 3 in our experiments, a value typically used for the GrIS (Aschwanden et al. 2012, Beckmann et al., 2023). "

- P4L28-29: Please check terminology on longitudinal stress and membrane stresses and revise sentence (cf. Hindmarsh, 2006, https://www.jstor.org/stable/25190296).

Adapted sentence:

Faster flowing ice, such as ice streams, glaciers, and shelves, is typically approximated using the SSA. In this case, longitudinal stretching dominates, and **membrane** and **transverse** stresses must be taken into account.

- P4L29-30: If you assume the SSA, you can speak of plug flow. The full ice column moves at the same horizontal speed.

We corrected the sentence as follows:

"The ice base is assumed to be slippery, and velocities at the bed equal velocities at the surface, allowing for depth averaging in the SSA equations."

Changed to:

"The ice base is assumed to be slippery, and the full ice column moved at the same horizontal speed. This plug flow allows for depth averaging in the SSA equation."

- P5L2: I would not use 'therefore' here. Just omit this adverb.

Thanks for your comment, we considered this correction in our manuscript.

- P5L6-7: Check consistency between q=0.6 and the Mohr-Coulomb sliding law as requested by the reviewer.

This section has been adapted as requested (changes in blood):

"The speed of basal sliding is determined by the sliding law, typically a power law relating to the basal shear stress $(\mathcal{T}b)$ and yield stress $(\mathcal{T}c)$. **We use an exponent of q = 0.6 in our "pseudo plastic" sliding law (Eq. 2)**.

$$\tau_b = -\tau_c \frac{\mathbf{u}}{u_{\text{threshold}}^q |\mathbf{u}|^{1-q}},$$

**To determine the yield stress $(\mathcal{T}c)$ we use the mohr-coulomb criterion in PISM.** The model considers basal resistance based on the hypothesis that the ice sheet rests on a till layer. The yield stress represents the strength of this aggregate material at the base of an ice sheet. When yield stress is lower than the driving stress $(\mathcal{T}c < \mathcal{T}d)$ there is likely to be sliding, and thus faster velocities can be observed. The driving stress in turn is dependent on the ice thickness (H) and surface gradients **(hs)** of the ice: $\mathcal{T}d \propto$ **H hs.** The thicker and steeper the ice, the higher the driving stress and most probably the ice velocity."

- P5L18-28: The symbology of Eq. 2 is confusing. Either correct or omit the section on the calving law and refer the interested reader to relevant literature. The article does not focus on the calving aspect anyway.

We adapted the equation in the main text as well as its description:

$$c = ||v|| \frac{\sigma}{\sigma_{max}}$$

where ||v|| is the velocity perpendicular to the ice front, σ is the von Mises stress for ice (Morlighem et al., 2016), and σmax is a threshold.

- P7L12: Please add to the paragraph that this spin-up ends again in the period 1961-1990 when the ice-sheet is assumed to have been in a quasi-equilibrium … (this was asked during the review and I think it is useful to mention the endpoint).

We completed the paragraph as follows (changes in blood):
"Starting in SIA-only mode, and an 18 km grid at -125 000 years, we refined our grid to 9 km at -25 000 years, and to 4.5 km at -5 000 years. For the last -1 000 years, we maintained a fixed resolution but introduced SSA to the SIA stress regime to represent the behavior of fast-flowing outlet glaciers. **Note that the initialisation of PISM ends after the reference period 1961-1990 when the ice sheet is assumed to have been in a quasi-equilibrium.**"

- Fig. 2: Omit the line from the left box 2 to the right box 1 in the lower right corner. It is redundant to the line from left box 1 to box 2.
-

Thanks for your comment, we considered this correction in Fig.2.

References

Aschwanden, A., Aðalgeirsdóttir, G., and Khroulev, C.: Hindcasting to measure ice sheet model sensitivity to initial states, The Cryosphere, 7, 1083–1093, https://doi.org/10.5194/tc-7-1083-2013, 2013.

Beckmann, J. and Winkelmann, R.: Effects of extreme melt events on ice flow and sea level rise of the Greenland Ice Sheet, The Cryosphere, 17, 3083–3099, https://doi.org/10.5194/tc-17-3083-2023, 2023.

---

## Author Response (AR4)

Technical comments of the paper "Coupling the regional climate MAR model with the ice sheet model PISM mitigates the melt-elevation positive feedback", by Delhasse et al.

Additional editor's private note:

LIST OF COMMENTS:

L114-115: Membrane stresses do comprise longitudinal stretching and transverse shearing. Please rephrase.
L115: Possibly rephrase: 'is assumed to move at the same horizontal speed'

Original sentences:

"Faster flowing ice, such as ice streams, glaciers, and shelves, is typically approximated using the SSA. In this case, longitudinal stretching dominates, and membrane and transverse stresses must be taken into account. The ice base is assumed to be slippery, and the full ice column moved at the same horizontal speed."

Are rephrased here:

"Faster flowing ice, such as ice streams, glaciers, and shelves, is typically approximated using the SSA. In this case, longitudinal stretching dominates, and membrane stresses must be taken into account. The ice base is assumed to be slippery, and horizontal speed is approximately equal throughout the depth of the ice."

L128: Capital letters for Mohr and Coulomb.

Ok thanks.

Fig. 2: Omit the line from the left box 2 to the right box 1 (2 --> 1) in the lower right corner. It is, in my view, redundant to the line from left box 1 to box 2. The same is valid for 9 --> 1. But you would then need a line from 1 --> 9.

Thank you for your comment. I guess you are requesting changes about Figure 1 and not Figure 2. We didn't well understand the comment in your last rapport and corrected by removing an arrow in the actual Figure 2. We then re-add this arrow in Figure 2, and finally correct the Figure 1 following this comment.